# Large Language Models as Topological Thinkers: A Benchmark on Graph Persistent Homology

## Abstract

Persistent homology offers a principled way to capture multi-scale topological structures in graphs, yet it remains unclear whether large language models (LLMs) can understand and reason about such high-order topological concepts. To address this gap, we introduce LLM4PH, the first benchmark designed to evaluate the ability of LLMs to comprehend and apply persistent homology on graphs. Our benchmark decomposes the persistent homology pipeline into four progressively challenging task levels, ranging from simplicial structure understanding to real-world graph inference. It includes 9 sub-tasks spanning 3 synthetic graph sizes and 3 real-world graph datasets, each annotated with topological features such as connected components, simplices, filtrations, and persistence diagrams. We systematically assess LLMs' capabilities in recognizing topological features, reasoning over filtrations, designing filtration strategies, and applying persistent homology for classification. Beyond task-level evaluation, we perform cross-task ablations on prompt encoding and transfer, explore post-training effects, and construct a compositional PH pipeline to assess end-to-end performance. Our results provide the first in-depth view of how well LLMs bridge discrete graph structures with continuous topological abstraction, and offer insights into their potential for structure-aware scientific reasoning.

## 1 Introduction

Large Language Models (LLMs) have demonstrated remarkable performance across a wide range of natural language understanding and reasoning tasks (Chang et al., 2024; Hadi et al., 2023; Plaat et al., 2024). However, their capabilities on graph-structured data, common in real-world, non-Euclidean domains, remain underexplored. Recent efforts have introduced benchmarks based on classical graph-theoretic problems such as shortest paths and connectivity, achieving notable progress on both static graphs (Fatemi et al., 2024; Wang et al., 2023) and dynamic graphs (Zhang et al., 2024). These benchmarks primarily focus on local structural reasoning and rely on standard node-edge representations. However, many real-world graphs exhibit complex high-order structures that go beyond pairwise interactions and require a more global topological perspective (Bick et al., 2023; Bianconi, 2021).

High-order structures refer to non-binary relationships among multiple nodes in a graph, such as communities in social networks or clusters in protein interaction networks. Some recent studies model these structures using graph patterns or hypergraphs and explore the use of LLMs for tasks like pattern matching (Dai et al., 2025; Feng et al., 2025). However, these representations often lack a natural sense of scale or continuity, making it difficult to analyze how structures evolve. Persistent homology(PH), a method from topological data analysis, offers an alternative by capturing multi-scale topological features in graphs (Pun et al., 2022). The process begins by assigning a filtration function, such as edge weights, which determines the order in which edges or nodes are added. PH then builds a sequence of simplicial complexes from this filtration and tracks the birth and death of topological features, resulting in a persistence diagram that summarizes their lifespan. This technique has been successfully applied in domains such as social network analysis, neuroscience, molecular biology, and materials science (Chen et al., 2022; Ye et al., 2023; Li et al., 2024; Curto & Sanderson, 2025; Obayashi et al., 2022).

For example, in a social network where edge weights reflect communication frequency, a filtration can be constructed by adding edges from the weakest to the strongest ties. As shown in Figure 1, graphs $G_a$ and $G_b$ differ only in the weight of a single edge, yet their topological structures evolve differently. Specifically, graph $G_a$ maintains a single connected component throughout the filtration, while graph $G_b$ initially has two separate components that eventually merge into one. PH captures these structural differences concisely, producing barcodes that summarize when components appear, persist, and merge across different interaction thresholds. This helps distinguish stable community structures from transient interactions and noise. For more detailed explanation of PH, please refer to Appendix C.

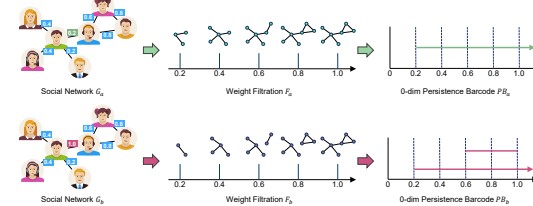

Figure 1: Social network filtration example.

Despite recent progress in benchmarking LLMs on graph-related tasks, many challenges remain in enabling language models to reason about graph data from a persistent homology (PH) perspective. From the graph side, this involves not just identifying structural patterns, but reasoning over filtration functions, edge activation thresholds, and multi-scale substructures that emerge over time. From the topological side, it demands understanding abstract mathematical notions such as simplicial complexes and homological features like connected components and cycles, and how these evolve across a sequence of filtered graphs. A model must not only understand the static shape of a graph but also track its dynamic topological evolution under varying filtrations.

To address these challenges, we introduce **LLM4PH**, a new benchmark designed to evaluate the capability of LLMs to understand, reason about, and apply PH to graph data. The benchmark decomposes the full PH workflow into interpretable sub-tasks, ranging from simple concepts like component counting and barcode reading to more complex reasoning such as filtration design, multi-step composition, and persistence-based graph comparison. We further include terminology ablation, compositional pipelines, and real-world graph settings to examine whether LLMs genuinely understand topological principles or rely on shallow pattern matching. Importantly, we also assess whether LLMs can serve as practical *assistants* in topological data analysis workflows—for example, by interpreting diagrams, selecting filtrations, or comparing barcodes in hybrid neuro-symbolic settings.

Through this structured evaluation, LLM4PH offers the first comprehensive testbed for investigating how well language models connect discrete graph structures with symbolic, algebraic, and topological abstraction.

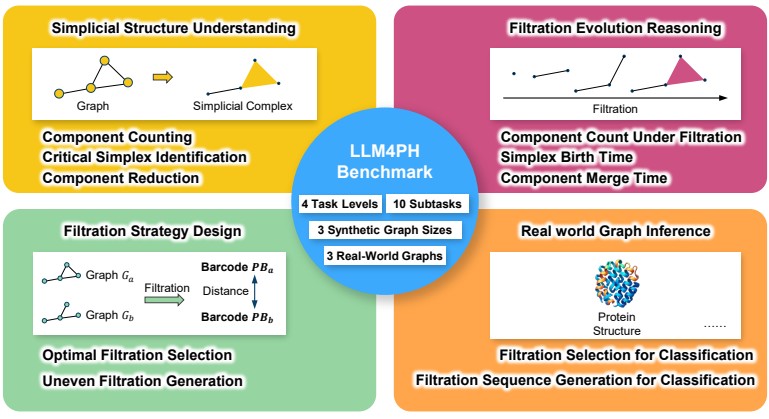

Figure 2: Benchmark Design.

## 2 BENCHMARK DESIGN

As shown in Figure 2, our benchmark is divided into four task categories, progressing from simple to complex with increasing difficulty. These tasks range from topological feature recognition to complex filtration strategy generation, and finally to practical applications, establishing a comprehensive

evaluation path from fundamentals to advanced concepts. The specific task designs and benchmark statistics are as follows:

## 2.1 TASK DESIGN

To rigorously assess the capability of LLMs in understanding PH on graphs, we design a benchmark composed of four progressively challenging task categories. In this benchmark, we employ the Vietoris–Rips (VR) complex, widely used in related research, as the simplicial complex constructed from graphs. For a comprehensive explanation of our task design methodology, see Appendix D.

**Simple Tasks (Simplicial Structure Understanding).** These tasks examine whether LLMs can recognize topological features such as connected components and critical simplices before the filtration. Tasks are evaluated by accuracy.

From the perspective of PH, these tasks correspond to the input stage before filtration begins, where LLM must identify the fundamental simplicial elements (e.g., 0-simplices and 1-simplices) that will later form the basis of the topological space. Accurate understanding at this level is essential for constructing suitable simplicial complexes.

- *0D Component Counting.* Counts the number of connected components (0-dimensional homology classes).
- *1D Simplex Counting.* Counts the number of 1-simplices (edges).
- *Component Reduction.* Modifies one edge in a given graph to reduce the number of connected components.
- *Persistence Diagram Interpretation.* Evaluate whether LLMs can correctly parse a PD represented in text and perform basic logical operations.
- *Topological Distance Reasoning.* Test whether LLMs can infer topological similarity between diagrams by reasoning over Wasserstein distances.

**Medium Tasks (Filtration Evolution Reasoning).** These tasks assess whether LLMs can reason about births and deaths of topological features. Tasks are evaluated by accuracy.

In the context of PH, these tasks correspond to tracking the dynamic evolution of topological features as the filtration progresses. The goal is to test whether LLMs can associate filtration steps with topological events such as component merging or cycle formation, which are essential for constructing persistence diagrams.

- *Simplex Birth Time.* Given a graph and its filtration sequence, predicts the birth time of specific simplices.
- *Component Merge Time.* Given a graph and its filtration sequence, predicts when specific connected components merge.
- *Component Count Under Filtration.* Given a graph and its filtration sequence, determines the number of connected components at a specified filtration value.

**Hard Tasks (Filtration Strategy Design).** These tasks challenge LLMs to select or generate filtration strategies that maximize the Wasserstein distance between the persistence barcodes of two given graphs. We consider two types of strategies: the first involves selecting a global filtration function, while the second involves generating a non-uniform sequence of filtration thresholds under a given filtration function. Tasks are evaluated using ranking-based metrics. From the PH perspective, this stage focuses on the design of filtration functions, which directly determines how simplicial complexes evolve and which topological features are revealed. LLMs are expected to reason backward from topological goals (e.g., maximizing feature differences) to suitable filtration strategies.

- *Optimal Filtration Selection.* Given two distinct graphs, the model selects the optimal filtration function from a set of candidates: node degree, edge weight, K-shell, closeness centrality, betweenness centrality, or eigenvector centrality, so as to maximize the Wasserstein distance between their resulting persistence barcodes.
- *Non-Uniform Filtration Generation.* Given two distinct graphs with edge weights ranging from 1 to 10, and a fixed filtration function (edge weight), the model generates a non-uniform filtration

sequence of length 5 (e.g., (1, 3, 4, 7, 10)) that maximizes the Wasserstein distance between their persistence barcodes.

**Real-World Tasks (Real-World Graph Inference).** These tasks evaluate the capability of LLMs to transfer and apply PH insights acquired from synthetic scenarios to real-world graph datasets. Specifically, we test whether LLMs can choose or generate suitable filtration strategies to solve real-world graph classification tasks. Tasks are evaluated by classification accuracy. The datasets are from Sutherland et al. (2003), corresponding to molecular graphs targeting benzodiazepine receptor (BZR), cyclooxygenase-2 (COX2), and dihydrofolate reductase (LDHFR), respectively.

This stage reflects the final step of the PH pipeline, where topological features are used for downstream tasks such as classification. It tests whether LLMs can operationalize abstract persistence information in practical settings.

- *Filtration Selection for Classification.* Given four graphs from two classes, the model selects the optimal filtration function from node degree, edge weight, K-shell, closeness centrality, betweenness centrality, or eigenvector centrality to group the graphs correctly by class.
- *Filtration Sequence Generation for Classification.* Given four graphs from two classes and a fixed filtration function with a set of candidate filtration values, the model generates a filtration sequence that enables correct classification of the four graphs into two groups.

Table 1: Overview of the LLM4PH Benchmark Tasks.

| Task Category | Task Name | Evaluation | #Samples |
|---|---|---|---|
| Simplicial Structure Understanding | Component Counting | Accuracy | 1200 |
| | Critical Simplex Identification | Accuracy | 1200 |
| | Component Reduction | Accuracy | 1200 |
| Filtration Evolution Reasoning | Component Count Under Filtration | Accuracy | 1200 |
| | Simplex Birth Time | Accuracy | 1200 |
| | Component Merge Time | Accuracy | 1200 |
| Filtration Strategy Design | Optimal Filtration Selection | Ranking | 1200 |
| | Uneven Filtration Generation | Ranking | 1200 |
| Real-World Graph Inference | Filtration Selection for Classification | Accuracy | 900 |
| | Filtration Sequence Generation for Classification | Accuracy | 900 |
| **Total** | 10 | | 11400 |

## 3 EXPERIMENTS

Based on the LLM4PH benchmark, we conduct a series of experiments to evaluate the performance of various large language models across different tasks. The goal is to assess whether LLMs are capable of understanding the abstract concepts behind PH and applying this knowledge to practical graph analysis tasks.

**Experimental Setup.** We evaluate the following six LLMs: GPT-4.1 (gpt-4.1-2025-04-14), GPT-4o (gpt-4o-2024-08-06), Claude(claude-3-7-sonnet-20250219), Gemini-2.5 (gemini-2.5-flash-preview-04-17), Deepseek-R1, Deepseek-V3, Qwen3-30B, Mistral-small-3.1-24B and Llama-3-70B. The first five are proprietary models accessed via their official APIs. The last three are open-source models deployed locally on a Linux server equipped with four NVIDIA A6000 GPUs. In some experiments, `Random` denotes predictions sampled uniformly from the empirical label distribution.

### 3.1 RESULTS OF SIMPLE TASKS (SIMPLICIAL STRUCTURE UNDERSTANDING)

**Performance analysis.** Simple tasks in our benchmark are conceptually related to prior LLM benchmarks on graphs that focus on counting connected components or identifying cycles Fatemi et al. (2024); Wang et al. (2023). However, instead of relying purely on graph-theoretic terminology, our benchmark frames these problems within the language of PH. For example, connected components are treated as 0-dimensional homology classes, and edges are seen as 1-simplices forming the foundation of higher-order topological features. This reframing not only allows for a more unified multi-scale perspective, but also tests whether LLMs can map familiar combinatorial structures into a topological reasoning framework.

Table 2: Results of Simple Tasks.

| Size | GPT-4.1 | GPT-4o | Claude | Gemini | DS-R1 | DS-V3 | Qwen | Mistral | Llama | Random |
|---|---|---|---|---|---|---|---|---|---|---|
| **0D Component Counting** | | | | | | | | | | |
| S | 0.693 | 0.863 | 0.995 | **1** | 0.995 | 0.998 | 0.995 | 0.843 | 0.390 | 0.301 |
| M | 0.518 | 0.733 | 0.915 | **1** | 0.915 | 0.995 | 0.978 | 0.680 | 0.335 | 0.303 |
| L | 0.375 | 0.375 | 0.528 | 0.988 | 0.528 | **0.998** | 0.535 | 0.455 | 0.335 | 0.353 |
| **1D Simplex Counting** | | | | | | | | | | |
| S | 0.213 | 0.270 | 0.150 | **0.271** | 0.150 | 0.260 | 0.143 | 0.085 | 0.223 | 0.253 |
| M | 0.105 | **0.225** | 0.068 | 0.078 | 0.068 | 0.043 | 0.035 | 0.030 | 0.058 | 0.255 |
| L | 0.105 | **0.178** | 0.018 | 0.044 | 0.018 | 0.038 | 0.020 | 0.045 | 0 | 0.265 |
| **Component Reduction** | | | | | | | | | | |
| S | **1** | 0.943 | **1** | **1** | **1** | **1** | 0.998 | 0.898 | 0.875 | - |
| M | 0.993 | 0.855 | 0.988 | **1** | 0.988 | **1** | 0.400 | 0.780 | 0.810 | - |
| L | 0.833 | 0.620 | 0.815 | **1** | 0.815 | **1** | 0.255 | 0.648 | 0.708 | - |

Table 3: Results of Medium Tasks.

| Size | GPT-4.1 | GPT-4o | Claude | Gemini | DS-R1 | DS-V3 | Qwen | Mistral | Llama | Random |
|---|---|---|---|---|---|---|---|---|---|---|
| **Simplex Birth Time** | | | | | | | | | | |
| S | 0.683 | 0.278 | 0.370 | 0.470 | **0.960** | 0.325 | 0.375 | 0.280 | 0.325 | 0.118 |
| M | 0.375 | 0.135 | 0.415 | 0.125 | **0.810** | 0.140 | 0.135 | 0.130 | 0.345 | 0.143 |
| L | 0.263 | 0.188 | 0.545 | 0.538 | **0.760** | 0.030 | 0 | 0.055 | 0.255 | 0.185 |
| **Component Merge Time** | | | | | | | | | | |
| S | 0.333 | 0.425 | 0.270 | 0.180 | **0.455** | 0.470 | 0.135 | 0.310 | 0.400 | 0.105 |
| M | 0.218 | **0.308** | 0.080 | 0.080 | 0.133 | 0.160 | 0.050 | 0.240 | 0.085 | 0.148 |
| L | **0.260** | 0.258 | 0.013 | 0.027 | 0.050 | 0.150 | 0.030 | 0.150 | 0.010 | 0.133 |
| **Component Count Under Filtration** | | | | | | | | | | |
| S | 0.968 | 0.790 | **0.990** | 0.885 | 0.988 | 0.838 | 0.535 | 0.515 | 0.448 | 0.173 |
| M | 0.958 | 0.593 | 0.885 | 0.810 | **0.995** | 0.660 | 0.448 | 0.422 | 0.240 | 0.120 |
| L | 0.615 | 0.210 | 0.510 | 0.635 | **0.975** | 0.335 | 0.285 | 0.130 | 0.113 | 0.252 |

As shown in Table 2, most LLMs perform very well on the *0D Feature Counting* task. Deepseek-V3 and Gemini reach near-perfect accuracy across all graph sizes, with open-source models such as deepseek-R1 and Qwen3-32B also achieving over 97% on small and medium graphs. This suggests that counting connected components is a well-internalized task for many models, especially when the concept is explicitly framed. Notably, the use of "0-dimensional homology" in the prompt does not significantly hinder performance, indicating that many models can align this term with the familiar notion of connectedness.

In contrast, the *1D Simplex Counting* task is considerably more challenging. Even the strongest models fail to surpass 30% accuracy on medium and large graphs. Among them, GPT-4o performs relatively better, but its results remain unsatisfactory. This difficulty likely stems from the inherently discrete nature of counting, as well as the weak association between topological terminology and natural language. Although the term "1-simplex" is mathematically precise, it is rarely used in everyday contexts, and most models may lack the prior knowledge needed to link it reliably to the concept of edges in graphs.

The *Connectivity Reduction* task sits between the two in difficulty. Most models achieve over 95% on small graphs, with Gemini maintaining strong performance even as the graph size increases. The task requires not only understanding the current number of connected components, but also modifying the graph to reduce it by one. This implies a degree of generative reasoning over topological state changes, and the performance gap between models such as Claude (81.5% on large graphs) and Qwen3 (25.5%) highlights their varying abilities to execute localized structural edits in a topological context.

**Observation.** While many LLMs perform well on classical graph tasks when expressed in plain language, only the strongest models sustain high accuracy when the same tasks are reframed using the abstract vocabulary of PH. **This reveals that true topological reasoning, understood as alignment with the PH pipeline rather than simple output matching, remains an unsolved challenge for most models.**

Table 4: Results of Hard Tasks. Subscripts indicate standard deviations.

| Size | GPT-4.1 | GPT-4o | Claude | Gemini | DS-R1 | DS-V3 | Qwen | Mistral | Llama |
|------|---------|--------|--------|--------|-------|-------|------|---------|-------|
| **Optimal Filtration Selection** | | | | | | | | | |
| S | $4.28_{1.75}$ | $4.38_{1.73}$ | $4.05_{1.79}$ | $4.06_{1.93}$ | $4.45_{1.77}$ | $4.22_{1.82}$ | $4.55_{1.73}$ | $3.67_{1.79}$ | $\mathbf{3.58}_{1.74}$ |
| M | $4.14_{1.50}$ | $4.18_{1.65}$ | $4.05_{1.61}$ | $4.38_{1.64}$ | $4.48_{1.52}$ | $4.30_{1.44}$ | $4.30_{1.66}$ | $3.78_{1.55}$ | $\mathbf{3.74}_{1.46}$ |
| L | $4.23_{1.40}$ | $4.24_{1.43}$ | $4.21_{1.43}$ | $4.40_{1.50}$ | $4.46_{1.48}$ | $4.20_{1.46}$ | $4.27_{1.60}$ | $\mathbf{4.05}_{1.49}$ | $4.10_{1.39}$ |
| **Non-Uniform Filtration Generation** | | | | | | | | | |
| S | $46.4_{30.1}$ | $56.0_{38.6}$ | $\mathbf{43.4}_{29.3}$ | $58.3_{34.2}$ | $60.2_{35.4}$ | $54.9_{34.6}$ | $53.7_{34.9}$ | $46.6_{31.5}$ | $50.8_{31.8}$ |
| M | $52.6_{35.1}$ | $54.2_{36.6}$ | $\mathbf{47.9}_{31.5}$ | $54.8_{35.8}$ | $64.4_{35.0}$ | $55.5_{33.5}$ | $59.1_{31.3}$ | $50.0_{33.3}$ | $53.9_{30.4}$ |
| L | $48.6_{33.9}$ | $53.0_{31.0}$ | $46.8_{30.3}$ | $54.7_{33.8}$ | $77.6_{33.5}$ | $54.0_{27.4}$ | $60.9_{34.2}$ | $\mathbf{46.7}_{31.1}$ | $52.6_{28.6}$ |

## 3.2 RESULTS OF MEDIUM TASKS (FILTRATION EVOLUTION REASONING)

**Performance analysis.** Unlike traditional graph reasoning tasks that rely solely on fixed structures, the medium tasks in our benchmark center around filtration. This dynamic process is foundational to PH, as it defines when features appear (birth) and disappear (death). In these tasks, models must go beyond static recognition and reason about how topological features evolve across a filtration sequence. This reflects one of the most distinctive aspects of PH and poses a significantly greater challenge for LLMs, especially without explicit computation.

As shown in Table 3, model performance varies widely across tasks and sizes, highlighting both the complexity of filtration-based reasoning and the limitations of current LLMs.

In the *Simplex Birth Time* task, which requires the model to determine when specific simplices (such as edges or triangles) appear during the filtration process, the results are highly inconsistent. DeepSeek-R1 achieves strong performance on this task (up to 96%), possibly due to its robustness in handling ordering-based reasoning. However, this advantage does not consistently extend to other tasks. In contrast, other open-source models perform poorly. For instance, Qwen reaches only 10% and Mistral just 5.5% on large graphs. These results suggest that reasoning about dynamic structural evolution remains a significant challenge for large language models.

The *Component Merge Time* task, which requires models to determine when two connected components merge during a filtration, is highly challenging for all models. Even top-performing models like GPT-4.1 and Claude score below 35%, and some models, such as Claude and Gemini on large graphs, barely produce any correct predictions. Although DeepSeek-R1 achieves the best results on small graphs, its performance declines significantly as the graph size increases. This difficulty further illustrates that tracking the merging of components across filtration steps is a fundamentally hard problem for autoregressive token-based large language models.

On *Component Count Under Filtration*, DS-R1 leads across all sizes with 98.8% on small, 99.5% on medium, and 97.5% on large, showing the strongest scale robustness. Claude is marginally higher on small at 99.0%, and GPT-4.1 stays strong on small and medium at 96.8% and 95.8% but drops to 61.5% on large. GPT-4o declines with size from 79.0% to 59.3% to 21.0%. DS-V3 sits mid tier. Open-source models remain far behind, near the mid 50s on small, mid 40s on medium, and below 30% on large, with Llama near 11.0%. Overall, DS-R1 sustains near-ceiling accuracy while others degrade with scale.

**Observation.** Filtration-based reasoning adds a distinct layer of complexity that exposes the structural limitations of current LLMs. While some models manage localized success on small graphs/simple filtrations, none consistently handle the full range of tasks and sizes. **PH introduces not only unfamiliar terminology but also a conceptual framework grounded in temporal evolution and geometric abstraction which remains difficult for most models to grasp.** These results pose a critical challenge in topological reasoning where continuity and change must be understood in tandem.

## 3.3 RESULTS OF HARD TASKS (FILTRATION STRATEGY DESIGN)

**Performance analysis.** This set of tasks evaluates LLMs' ability to reason in reverse: from desired objectives (e.g., maximizing topological difference) to designing appropriate filtration strategies. In contrast to earlier tasks that test a model's ability to recognize topological features, these tasks assess whether a model can actively control the filtration process to induce specific topological outcomes.

We use ranking-based evaluation for both tasks. In *Optimal Filtration Selection*, each model is asked to choose a filtration function (e.g., node degree, edge weight) that maximizes the Wasserstein distance between persistence barcodes of two input graphs. Since there are six candidate functions, the model's answer is ranked among the six: lower rank means better performance (1 is best, 6 is worst). In *Non-Uniform Filtration Generation*, the model generates a filtration sequence (a length-5 subset from values 1 to 10), and its output is ranked among 126 possible sequences based on the resulting Wasserstein distance. Again, lower rank indicates a more effective choice.

From Table 4, we observe that all models struggle with these tasks. In the *Optimal Filtration Selection* task, the small open-source models Mistral and Llama surprisingly achieve the best results, while most other models fail to produce any rankings below 3. In addition, all models exhibit large standard deviations, indicating highly unstable outputs. This suggests that their performance is only marginally better than random guessing. Overall, designing filtrations based on specific goals and graph structures remains a highly challenging task for current language models.

In the *Non-Uniform Filtration Generation* task, Claude emerges as the best-performing model across all graph sizes, achieving the lowest average rank (43.4 on small, 47.9 on medium, and 46.8 on large). This indicates a surprising strength in generating effective filtration sequences that yield topological divergence. However, the overall performance remains disappointing. All models exhibit large standard deviations that are nearly as large as their means, suggesting that the models have not truly learned any meaningful knowledge related to the task.

**Observation.** Topological control tasks remain challenging for LLMs, requiring reasoning about graph-filtration interactions. Our benchmark tests active topological shaping rather than passive interpretation. Results show that even strong models lack sufficient reasoning granularity, suggesting new research directions in persistence-aware training and hybrid architectures. **Success requires aligning structural patterns with topological objectives, not just pattern recognition.**

## 3.4 RESULTS OF REAL-WORLD TASKS (OR REAL-WORLD GRAPH INFERENCE)

Table 5: Results of Real-World Tasks.

| Datasets | GPT-4.1 | GPT-4o | Claude | Gemini | DS-R1 | DS-V3 | Qwen | Mistral | Llama |
|---|---|---|---|---|---|---|---|---|---|
| **Filtration Selection for Classification** | | | | | | | | | |
| BZR | **0.350** | **0.350** | **0.350** | 0.340 | 0.320 | **0.350** | 0.320 | **0.350** | **0.350** |
| COX2 | 0.320 | 0.260 | 0.320 | 0.320 | **0.360** | 0.320 | **0.360** | 0.290 | 0.315 |
| LDHFR | 0.345 | 0.310 | 0.345 | 0.345 | 0.460 | 0.335 | **0.460** | 0.310 | 0.340 |
| **Filtration Sequence Generation for Classification** | | | | | | | | | |
| BZR | 0.595 | 0.650 | **0.775** | 0.765 | 0.760 | 0.410 | 0.400 | 0.652 | 0.740 |
| COX2 | 0.710 | 0.555 | 0.695 | **0.755** | 0.180 | 0.160 | 0.480 | 0.655 | 0.655 |
| LDHFR | 0.950 | 0.920 | **0.970** | 0.958 | 0.760 | 0.880 | 0.740 | 0.860 | 0.945 |
| **Directly Classification** | | | | | | | | | |
| BZR | **0.985** | 0.940 | 0.990 | 0.969 | 0.920 | 0.980 | 0.820 | 0.525 | 0.995 |
| COX2 | 0.985 | 0.920 | 0.995 | 0.990 | 1 | 0.995 | 0.740 | 0.460 | 1 |
| LDHFR | 1 | 0.960 | 0.995 | 1 | 1 | 0.990 | 0.980 | 0.790 | 1 |

This set of tasks evaluates whether LLMs can apply PH to graph classification. It includes two PH-guided tasks: *Filtration Selection for Classification* and *Filtration Sequence Generation for Classification*, as well as a direct classification baseline without topological reasoning.

In the *Filtration Selection for Classification* task, performance is notably poor and highly uniform across models and datasets, with accuracies fluctuating narrowly around 0.32 to 0.36. This indicates that models consistently choose similar filtration functions regardless of the graph structure. Further inspection reveals that most LLMs default to selecting `edge weight` as the filtration function, possibly because it appears more "interpretable" or statistically grounded. However, this uniformity fails to expose discriminative topological differences, leading to almost random classification results.

In contrast, the *Filtration Sequence Generation for Classification* task exhibits stronger performance across all models, especially on the LDHFR dataset, where top models (Claude, GPT-4.1, Gemini) achieve accuracies above 0.95. This suggests that when explicitly instructed to manipulate filtration sequences, even under fixed filtration functions, LLMs can induce more topologically meaningful

separations. Notably, Claude outperforms all other models on BZR and LDHFR, achieving the best classification accuracy among all methods.

To evaluate the reliance on PH reasoning, we introduce a control task: *Direct Classification*, where the model is asked to classify graphs directly without designing filtration strategies. Surprisingly, most models achieve near-perfect accuracy (0.98–1.0), suggesting that for small 4-graph inputs, LLMs can leverage shallow statistical patterns or implicit textual correlations to memorize or infer class labels. However, this also reveals that the PH tasks are substantially more difficult than the direct version, requiring abstraction, planning, and topological insight rather than pattern matching.

**Observation.** The weak performance on filtration function selection, contrasted with stronger results on filtration sequence generation, suggests that LLMs are capable of expressing meaningful topological reasoning **only when guided by structured prompts and constrained options**. In the absence of such scaffolding, models tend to fall back on familiar heuristics, which often obscure rather than uncover PH-based structure.

### 3.5 CROSS-TASK ANALYSES

Beyond the evaluation of individual tasks, we conduct three cross-task experiments that focus on broader aspects of model behavior. These experiments examine the effects of input representation, model fine-tuning, and pipeline compositionality across multiple tasks and graph sizes. Together, they provide additional insight into the generalization and operational behavior of language models when applied to PH reasoning. Detailed results for all cross-task experiments are provided in Appendix E.

**Prompt Encoding and Task Formulation Ablation** We examine how the structure of prompts influences model behavior across tasks. This experiment includes variations in both the **graph representation style** and the **task formulation style**, applied to three representative tasks: 1D Simplex Counting, Component Merge Time, and Component Count Under Filtration on both synthetic and real-world LDHFR data. We evaluate three styles of graph representation, i.e., text-style format, code-like format, and matrix-based format, and three styles of task instruction formulation: topological, graph-theoretic, and minimal. Descriptions of each style are summarized in Table 12, and examples for each combination are provided in the appendix. From Table 8, text-style input consistently yields the highest accuracy across models. Graph-theoretic task phrasing also outperforms both topological and minimal formulations in most settings. In contrast, matrix-based input and minimal task prompts tend to result in lower and less stable performance. These findings confirm that surface-level prompt features play a substantial role in model outcomes, even when the underlying graph structure and reasoning objectives remain unchanged. Consequently, evaluations of language models on topological reasoning must account for prompt formulation as a key factor in performance variance.

To further investigate the effect of task formulation, we conduct a focused **terminology ablation study**. For each task, we generate paired prompts with topological phrasing (e.g., H0 class) and graph-theoretic equivalents (e.g., connected component"), keeping the underlying operations identical. Results in Table 9 show that terminology sensitivity is task-dependent: for example, GPT-4o achieves similar accuracy across formulations on the birth" task, but shows a significant drop in the CCUF" task when PH-specific terminology is used (from 0.8633 to 0.45). This indicates that while domain terminology contributes to performance variance, it does not fully explain failure modes—many errors stem from deeper structural reasoning limitations rather than surface-level linguistic mismatches.

To evaluate whether prompt engineering can mitigate reasoning errors in complex settings, we conduct an experiment on the non-uniform filtration generation task using two prompting strategies: Few-shot Chain-of-Thought (CoT) and rule-emphasized prompting. While the former provides a complete reasoning example to encourage mimicking, the latter enforces strict filtration and topology rules without examples. As shown in Table 10, neither approach yields consistent improvement. Performance remains unstable, with large variance across both models. This suggests that prompt design alone cannot resolve the deeper challenge of maintaining accurate H0/H1 transitions in multi-step topological reasoning.

**Post Training and Scale Transferability** We study whether supervised adaptation can instill stable topological routines in an open-source model. We fine tune LLaMA-3.1-8B-Instruct on small graphs for three representative tasks and then evaluate on medium graphs with matched distributions. The protocol preserves the prompt settings used in the ablation, which allows us to measure how much

performance remains sensitive to surface representation after adaptation. As reported in Table 11, post training delivers consistent and often large gains across all tasks. On 1D Simplex Counting the best prompt improves from 0.02 to 0.20, while Code like with topological phrasing reaches 0.26. On Component Merge Time the score rises from 0.09 to 0.43 for text with graph theoretic phrasing, with similar improvements for other text prompts. On Component Count Under Filtration the text with graph theoretic phrasing increases from 0.16 to 0.66, and code like with topological phrasing reaches 0.54. Prompt sensitivity remains present after adaptation. Text prompts continue to be reliable across tasks. Code like prompts become more competitive after fine tuning and show the largest relative improvements in two tasks. Minimal phrasing improves but still trails behind explicit instructions. We observe the same trends in cross scale transfer, where a model tuned on small graphs improves from 0.005 to 0.235 on large 1D Simplex Counting, which suggests that the learned routines generalize beyond the training size. Overall, fine tuning strengthens topological reasoning and reduces variance, yet careful prompt design remains important for peak performance.

**Compositional PH Pipeline** Beyond evaluating LLM performance on individual subtasks, we construct a full pipeline to assess how well LLMs perform when guiding an integrated PH workflow. We run a *Filtration Selection for Classification* experiment on the real-world BZR dataset. The pipeline follows the standard five-step process used by PH practitioners: (1) select a filtration function, (2) compute filtration values, (3) generate persistence diagrams, (4) compare diagram distances, and (5) determine the final class label. We also include results obtained directly from persistent homology tools, using four filtration functions: Weight, Degree, Betweenness, and Closeness.

In Table 6, `LLM` indicates that a step is completed entirely by the language model through prompt-based reasoning, while `code` denotes that the step is handled by standard PH libraries such as `Gudhi`. We test four settings that vary the boundary between model reasoning and code execution. In Setting 1 the model selects the filtration and also executes diagram generation and distance comparison, while code handles filtration values and the final classifier. In Setting 2 the model selects the filtration and all downstream steps use code. In Setting 3 the model selects the filtration and performs distance comparison, while code generates the diagrams. In Setting 4 the model selects the filtration and generates diagrams, while code performs distance comparison and the final classifier.

Table 6 shows that the pure code-based configuration (filtration by the model, all downstream steps by code) produces stable mid-level accuracy across models, providing a consistent baseline. In contrast, hybrid designs where the model participates in distance comparison achieve the highest scores for Claude and DS-V3. Settings that require the model to generate diagrams are less reliable, often leading to performance drops. These results suggest that the language model is most effective in filtration selection and, for some models, in distance reasoning, while code remains essential for diagram generation and overall stability. The results obtained directly from persistent homology tools show that although PH tool based pipelines consistently achieve higher end to end accuracy than LLM driven configurations, hybrid pipelines can occasionally match or even surpass tool only performance in specific settings. For instance, Claude reaches an accuracy of 0.840 when combined with symbolic components. These findings illustrate the potential of hybrid reasoning loops and highlight particular interfaces, such as filtration selection, where LLMs can provide meaningful contributions.

**Large-Graph Scalability via Subgraph Sampling** Since current context windows cannot contain full large graphs, we evaluate scalability through controlled subgraph sampling on the ego-Facebook dataset Leskovec & Mcauley (2012). We use random walk sampling to build six collections with 10, 20, 50, 80, 100, and 150 subgraphs, each subgraph containing 50 nodes. We run the 1D Simplex Counting task five times for GPT-4o and GPT-4.1-mini and report mean accuracy (standard deviation) in Table 7. Both models improve as the number of sampled subgraphs grows. GPT-4.1-mini increases monotonically from 14% to 24%, and GPT-4o shows an overall upward trend.

**Error analysis: Non-Uniform Filtration Generation** To better understand model failure on this task, we conduct a systematic error analysis and identify six recurring patterns for the Non-Uniform Filtration Generation task. These fall into three broad categories: loss of prior context, incomplete procedural reasoning, and chain-of-thought instability.

1. *Fact forgetting.* The model fails to retain previously mentioned edge information. For instance, it may first note that node 1 is connected to node 7, but later treat them as belonging to separate components. This occurs because earlier facts drift out of the model's attention window and are no longer integrated into downstream predictions.

2. *Rule forgetting.* After explicitly acknowledging rules, such as "isolated nodes are not counted as components" or "a merge occurs when two components join". This reflects the absence of persistent logical constraints; once the model's focus shifts, prior rules are no longer enforced.

3. *Incomplete algorithm emulation.* The model often attempts to reproduce the output of traversal-based algorithms (such as depth-first search) but does so heuristically, without internal state tracking. This results in invalid outputs, such as calling a walk with repeated nodes a cycle or skipping key edges during merge detection.

4. *Blurry conceptual boundaries.* Topological concepts are applied too loosely. For example, any path resembling a loop may be labeled a "hole" or "cycle," even if it includes repeated vertices or fails basic structural criteria. This suggests the model relies on surface-level statistical cues rather than precise graph-theoretic definitions.

5. *Interrupted reasoning.* In longer examples requiring multiple stages of analysis, the model's output may terminate mid-thought (e.g., "Now we check for cycles…" with no continuation). This typically occurs when the chain of thought becomes too long to maintain internal coherence, especially in graphs with many nodes or edge weights.

6. *Analysis paralysis.* When faced with multiple plausible merge paths or interpretations, the model may enter loops of self-revision. For instance, it might state "the correct answer is 4.00," then immediately consider "perhaps 10.00," and continue revising. This failure to converge indicates difficulty in managing uncertainty and resolving conflicting hypotheses.

These failure modes reveal structural limitations in current LMs. Robust reasoning under non-uniform filtrations needs persistent memory, explicit constraint handling, and the ability to simulate structured procedures.

## 4 CONCLUSION

This work introduces LLM4PH, the first benchmark designed to evaluate large language models' ability to understand, reason about, and apply PH on graphs. By decomposing the PH pipeline into interpretable sub-tasks and organizing them into a progression of difficulty, we assess both the conceptual alignment and reasoning depth of LLMs in a topological setting. Our results reveal that while some models can handle low-dimensional features or structured prompts, most struggle with dynamic evolution and filtration design, indicating a fundamental gap between current LLM capabilities and the demands of topological abstraction. We hope this benchmark provides a foundation for future advances in topology-aware reasoning, hybrid neuro-symbolic architectures, and geometric representation learning with language models. Improving LLMs' ability to engage with topological structures could unlock new directions in scientific discovery, graph-based inference, and interpretable AI systems grounded in geometric reasoning.

**Ethics Statement**   This study does not involve human subjects, sensitive personal data, or any proprietary or confidential information. All datasets used in this work are synthetic or publicly available and are pre-processed to remove any personally identifiable information. For real-world graph datasets, we follow standard usage and citation practices and do not modify the underlying node or edge semantics.

**Reproducibility Statement**   To ensure full reproducibility, we provide all datasets, prompts, task definitions, and evaluation scripts in an anonymous GitHub repository: `https://anonymous.4open.science/r/LLM4PH-7015/`. Detailed descriptions of each benchmark task, including data generation methods, prompt templates, and scoring metrics, are included in the main paper and the appendix.

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

## THE USE OF LARGE LANGUAGE MODELS (LLMS)

We used large language models (LLMs), such as ChatGPT, to assist with writing refinement, code debugging, and literature exploration. LLMs helped improve the clarity and academic tone of the manuscript, assisted in refactoring evaluation scripts, and supported the search for relevant background references. All core research ideas, experimental design, implementation, and analysis were conducted independently by the authors. No text or results were generated solely by LLMs.

## A  RELATED WORK

**LLM for Graphs.**   Recent efforts have sought to evaluate the reasoning capabilities of large language models (LLMs) on graph-structured data through benchmark datasets. Early benchmarks focus on static graph tasks such as shortest path computation, cycle detection, node classification, and graph traversal Fatemi et al. (2024); Wang et al. (2023). More recent work expands this scope to dynamic graphs, introducing temporal reasoning tasks such as dynamic link prediction and event ordering Zhang et al. (2024). Other studies have explored graph pattern recognition Dai et al. (2025) and subgraph isomorphism through natural language prompts. These benchmarks typically emphasize local structure, edge-level logic, or symbolic reasoning over discrete structures. However, they often overlook the global, multi-scale, and high-dimensional characteristics inherent in many real-world graphs. Our work complements and extends this line of research by introducing PH as a testbed for evaluating topological reasoning beyond pairwise relationships.

Recent work has explored using LLMs for traditional graph learning tasks Chen et al. (2024b). Methods like Graph-LLM Chai et al. (2023) and GraphText Zhao et al. (2024) convert graphs into natural language for few-shot inference, while others combine LLMs with GNNs Tang et al. (2024); Tan et al. (2024). However, these approaches focus on low-order features rather than high-dimensional topological structures. Our work complements this by targeting PH features and assessing LLMs' ability to reason about them without neural encoders.

**Persistent Homology on Graphs.**   Persistent homology, a core tool in topological data analysis (TDA), provides a framework to extract multi-scale topological features from data (Pun et al., 2022). When applied to graphs, it captures global structures such as connected components, cycles, and voids across a filtration of the graph Aktas et al. (2019); Immonen et al. (2023); Horak et al. (2009). Graph-based persistent homology has been widely used in fields like neuroscience Dabaghian et al. (2012); Curto & Sanderson (2025), biology Xia & Wei (2014); Meng et al. (2020); Townsend et al. (2020), and materials science Obayashi et al. (2022), often serving as a shape descriptor or structural feature for downstream tasks. In machine learning, persistent diagrams have been integrated into graph neural networks Hofer et al. (2019); Yan et al. (2021); Chen et al. (2024a) as interpretable features. Despite its growing adoption, prior work assumes full access to the topological pipeline and does not explore whether LLMs can internalize or reason over such structures directly. Our work bridges this gap by proposing a benchmark that tests whether LLMs can understand, predict, and generate persistent homology concepts from graph inputs without explicit computation.

**Simplicial Complex Identification**   Recent efforts in understanding and identifying higher-order topological structures have extended classical graph methods to the realm of hypergraphs and simplicial complexes Babai & Codenotti (2008) analyzed the complexity of hypergraph isomorphism, showing that while the general problem is NP-complete, isomorphism between bounded-rank hypergraphs, such as low-dimensional simplicial complexes, admits moderately exponential-time algorithms via group-theoretic methods. To capture structural similarity beyond exact isomorphism, recent work has introduced polynomial-time heuristics based on combinatorial refinement. Feng et al. (2024) proposed a generalization of the Weisfeiler–Lehman (WL) test to hypergraphs and constructed WL-based hypergraph kernels, enabling efficient comparisons between high-order structures through subtree and hyperedge patterns. Building on this, Zhang et al. (2025) reinterpreted such kernels in terms of homomorphism counts, revealing that many existing methods implicitly focus on acyclic patterns and introducing the Subtree-Cycle Kernel to enhance expressiveness by capturing cyclic features. These approaches provide both theoretical insights and practical tools for

identifying and comparing simplicial complexes through a combination of isomorphism heuristics and homomorphism-informed embeddings.

## B    DATASET CONSTRUCTION

To support a controlled and interpretable evaluation of persistent homology understanding, we construct both synthetic and real-world datasets tailored to each task category.

### B.1    BENCHMARK STATISTICS

The statistical overview of our benchmark is illustrated in Table 1. Each synthetic dataset task category's samples are composed of an equal number of large, medium, and small graphs (10 nodes, 15 nodes, 30 nodes). For example, out of 1200 samples, 400 are large graphs, 400 are medium graphs, and 400 are small graphs. The real-world datasets BZR, COX2, and LDHFR have an equal number of samples. For more detailed information about dataset construction, please refer to Appendix B.

### B.2    SYNTHETIC GRAPH GENERATION

All synthetic graphs are generated using randomized network models with fixed size constraints. Specifically, small, medium, and large graphs are set to contain 10, 15, and 30 nodes respectively. For each graph, we randomly sample edges to ensure structural variability while maintaining connectivity and topological relevance.

### B.3    TASK-SPECIFIC FILTERING AND SELECTION

For **simple** and **medium** tasks, we compute the Vietoris–Rips complex from each candidate graph and evaluate its corresponding Betti numbers and persistence diagrams across standard filtration functions. Graphs are selected based on whether they yield diverse and interpretable topological outcomes, such as distinct connected components, nontrivial 1-dimensional features, and clear merging events.

For **hard tasks**, we compute pairwise Wasserstein distances between persistence diagrams of graph pairs under all candidate filtration strategies. Each strategy (e.g., node degree, edge weight, closeness centrality) is evaluated, and we retain the ranking of each strategy based on the resulting distance. This ranking provides the ground truth supervision for tasks involving filtration design.

### B.4    REAL-WORLD GRAPH SELECTION

For **real-world tasks**, we curate graphs from public datasets (e.g., BZR, COX2, LDHFR) and sample small subsets of graphs from two distinct classes. For each problem instance, we evaluate the Wasserstein distance between diagrams produced under each candidate filtration strategy, and label tasks according to whether the correct class separation can be achieved. Graphs are allowed to appear in at most two tasks to avoid overfitting and ensure evaluation diversity.

This construction pipeline ensures that each benchmark task is grounded in verifiable topological variation and aligned with persistent homology computation standards.

## C    ADDITIONAL EXPLANATION OF PERSISTENT HOMOLOGY

Persistent homology is a core method in topological data analysis (TDA) that enables the extraction of multi-scale topological features from data. It generalizes classical homology theory by introducing the concept of a *filtration*, which allows one to track how topological features evolve across scales.

A filtration is a nested sequence of simplicial complexes:

$$K_0 \subseteq K_1 \subseteq \cdots \subseteq K_T,$$

typically constructed by gradually adding simplices based on a filtration function (e.g., edge weights in a graph). As the filtration evolves, features such as connected components (0-dimensional homology), cycles (1-dimensional), and voids (2-dimensional) appear and eventually disappear.

Key terms used throughout this work include:

- **Simplex**: A generalization of a vertex (0-simplex), edge (1-simplex), triangle (2-simplex), or higher-dimensional face. Simplices are the building blocks of a simplicial complex.

- **Simplicial Complex**: A finite set of simplices closed under the subset operation. It defines a discrete topological space amenable to homology computation.

- **Filtration Function**: A function $f$ that assigns a real-valued threshold to each simplex (often indirectly through edges or vertices). The function governs the order in which simplices are added during the filtration.

- **Birth and Death**: A topological feature (e.g., a connected component or a cycle) *births* at the filtration index where it first appears and *dies* when it is merged into a larger feature or filled in.

- **Persistence Diagram / Barcode**: A multiset of intervals $\{[b_i, d_i]\}$ representing the lifespan of topological features. The length $d_i - b_i$ reflects the *persistence* or significance of each feature.

- **Betti Number** $\beta_k$: The rank of the $k$-th homology group $H_k$, counting $k$-dimensional holes: $\beta_0$ for components, $\beta_1$ for loops, etc.

- **Wasserstein Distance**: A distance metric between persistence diagrams that quantifies how topological structures differ across filtrations. It is often used to evaluate the output of persistence-aware models.

By computing and analyzing these topological signatures, persistent homology provides a robust, noise-tolerant summary of data structure that is invariant to continuous transformations and particularly suited to graph-structured domains.

## D    DETAILS OF TASK DESIGN

This appendix provides detailed explanations for each task in the LLM4PH benchmark, framed through the mathematical language of persistent homology and simplicial complex theory.

### D.1    SIMPLE TASKS (SIMPLICIAL STRUCTURE UNDERSTANDING)

These tasks correspond to the initial stage of persistent homology, prior to any filtration, where the simplicial structure of the graph must be identified. Each task tests whether an LLM can understand basic homological objects from a combinatorial graph input.

**0D Component Counting.**    Given an undirected graph $G = (V, E)$, the model is asked to compute $\beta_0(G)$, the 0-th Betti number, which equals the number of connected components in the graph. This is equivalent to computing the rank of the 0-dimensional homology group $H_0(G)$.

**1D Simplex Counting.**    Given a graph $G$, the task requires counting the number of 1-simplices, which correspond to edges in the 1-skeleton of the simplicial complex induced by $G$. This measures the cardinality of the 1-simplex set $\Sigma_1$.

**Component Reduction.**    Given $G$, the model is asked to identify a single edge $e \notin E$ such that $E' = E \cup \{e\}$ decreases $\beta_0(G)$ by one. This requires reasoning about connected component merges via edge addition.

### D.2    MEDIUM TASKS (FILTRATION EVOLUTION REASONING)

These tasks test an LLM's ability to reason over a filtration $\{K_t\}_{t \in T}$ of simplicial complexes built from a weighted graph $G$, where $K_t$ denotes the simplicial complex at filtration threshold $t$. The aim is to understand topological evolution: the birth and death of homology classes across $t$.

**Simplex Birth Time.** Given a weighted graph $G$ and a filtration order (e.g., by edge weight), the model predicts the value $t$ at which a given simplex $\sigma \in K_t$ appears. This corresponds to the birth time $b(\sigma)$ in the persistence diagram.

**Component Merge Time.** For a pair of initially disconnected vertices $(u, v)$, the model predicts the threshold $t$ at which $u$ and $v$ are first included in the same connected component, i.e., when their representatives in $H_0$ are merged.

**Component Count Under Filtration.** At a given threshold $t$, the model is asked to compute $\beta_0(K_t)$, the number of connected components in the complex $K_t$.

### D.3 HARD TASKS (FILTRATION STRATEGY DESIGN)

These tasks involve reasoning from desired topological divergence toward filtration design. Given two graphs $G_1$ and $G_2$, the objective is to select or generate a filtration function $f : E \to \mathbb{R}$ (or a sequence of filtration thresholds) that induces persistence diagrams $D_1, D_2$ with maximal Wasserstein distance $W(D_1, D_2)$.

**Optimal Filtration Selection.** The model selects a global filtration function $f \in \mathcal{F}$ from a predefined set (e.g., node degree, edge weight, K-shell, etc.) such that the resulting persistence diagrams maximize $W_p(D_1, D_2)$ under a $p$-Wasserstein metric.

**Non-Uniform Filtration Generation.** Under a fixed filtration function $f$, the model selects a non-uniform filtration sequence $\{t_1, t_2, \ldots, t_5\}$ from a discrete set (e.g., $\{1, 2, \ldots, 10\}$) to maximize $W(D_1, D_2)$, effectively constructing a coarse but topologically expressive filtration.

### D.4 REAL-WORLD TASKS (REAL-WORLD GRAPH INFERENCE)

These tasks examine whether an LLM can apply persistent homology insights to downstream tasks such as graph classification. Let $\mathcal{G} = \{G_1, G_2, G_3, G_4\}$ be a set of graphs belonging to two classes. The goal is to design a filtration such that the resulting persistence-based representations $\{D_i\}$ allow for correct class separation.

**Filtration Selection for Classification.** The model selects a filtration function $f \in \mathcal{F}$ that yields persistence diagrams best aligned with the class partition of the graphs, typically optimizing for inter-class distance or clustering.

**Filtration Sequence Generation for Classification.** Under a given filtration function $f$, the model generates a filtration sequence $\{t_1, \ldots, t_5\}$ that maximizes topological separability between graphs from different classes, typically measured by diagram-level distance or homology-aware clustering accuracy.

## E SUPPLEMENTARY TABLES FOR CROSS-TASK ANALYSES

This appendix provides supplemental materials to support the results presented in the Sec. 3.5. We include:

- **Prompt Style Summary (Table 12)**, outlining the different ways graph structure and task instructions are encoded.
- **Prompt Ablation Results (Table 8)**, showing performance across multiple representation and instruction combinations.
- **Post-Training Evaluation (Table 11)**, reporting transfer results after supervised tuning on small graphs.
- **Compositional Pipeline Evaluation (Table 6)**, examining model performance in hybrid workflows across persistent homology steps.

- **Subgraph Scaling (Table 7)**, assessing accuracy trends as the number of sampled subgraphs increases.

Table 6: Compositional PH pipeline on BZR. The model chooses the filtration in every row.

| Settings | | | | | Accuracy | | | | |
|---|---|---|---|---|---|---|---|---|---|
| Filtration selection | Filtration values | Diagram generation | Distance comparison | Classifier | GPT-4o | GPT-4.1 | Claude | DS-V3 | Mistral |
| LLM | CODE | LLM | LLM | CODE | 0.150 | 0.180 | 0.780 | 0.290 | 0.210 |
| LLM | CODE | CODE | CODE | CODE | 0.350 | 0.350 | 0.350 | 0.350 | 0.350 |
| LLM | CODE | CODE | LLM | CODE | 0.250 | 0.280 | 0.840 | 0.780 | 0.520 |
| LLM | CODE | LLM | CODE | CODE | 0.300 | 0.320 | 0.190 | 0.025 | 0.150 |
| WEIGHT | CODE | CODE | CODE | CODE | | | 0.790 | | |
| DEGREE | CODE | CODE | CODE | CODE | | | 0.610 | | |
| BETWEENNESS | CODE | CODE | CODE | CODE | | | 0.890 | | |
| CLOSENESS | CODE | CODE | CODE | CODE | | | 0.800 | | |

Table 7: Subgraph sampling on ego–Facebook for 1D Simplex Counting. Values are mean accuracy $\pm$ standard deviation over five runs.

| Subsamples | 10 | 20 | 50 | 80 | 100 | 150 |
|---|---|---|---|---|---|---|
| GPT–4o | 0.0600±0.08 | 0.1400±0.08 | 0.1280±0.04 | 0.1325±0.03 | 0.1733±0.02 | 0.1960±0.02 |
| GPT–4.1–mini | 0.1400±0.10 | 0.1400±0.08 | 0.1520±0.03 | 0.1850±0.03 | 0.2260±0.02 | 0.2413±0.04 |

Table 8: Prompt encoding and task formulation ablation shown in a single horizontally merged table across three tasks.

| Prompt setting | 1D Simplex Counting | | | | Component Merge Time | | | | Component Count Under Filtration | | | |
|---|---|---|---|---|---|---|---|---|---|---|---|---|
| | Synthetic | | LDHFR | | Synthetic | | LDHFR | | Synthetic | | LDHFR | |
| | GPT-4o | Claude | GPT-4o | Claude | GPT-4o | Claude | GPT-4o | Claude | GPT-4o | Claude | GPT-4o | Claude |
| Text style + Graph theoretic | **0.270** | 0.150 | 0.170 | 0.260 | 0.425 | 0.270 | **0.250** | 0.090 | 0.785 | **0.983** | 0.173 | **0.780** |
| Text style + Minimal | 0.176 | 0.078 | 0.273 | **0.670** | 0.443 | 0.225 | 0.160 | 0.023 | 0.093 | 0.025 | 0.233 | 0.085 |
| Text style + Topological | 0.160 | 0.268 | 0.250 | 0.250 | **0.510** | 0.090 | 0.125 | 0.020 | 0.633 | 0.790 | 0.148 | 0.655 |
| Matrix style + Topological | 0.125 | 0.224 | 0.118 | 0.230 | 0.245 | 0.090 | 0.018 | 0.000 | 0.643 | 0.870 | 0.180 | 0.610 |
| Code like + Topological | 0.128 | 0.210 | 0.118 | 0.160 | 0.445 | 0.115 | 0.220 | 0.110 | 0.833 | 0.980 | 0.120 | 0.570 |

Table 9: Terminology ablation results comparing topological vs. graph theoretic phrasing across three tasks.

| Condition | GPT 4.1 mini | GPT 4o |
|---|---|---|
| Graph terminology birth | 0.67 | 0.33 |
| Topology terminology birth | 0.6533 | 0.3233 |
| Graph terminology merge | 0.02 | 0.5533 |
| Topology terminology merge | 0.0066 | 0.57 |
| Graph terminology CCUF | 0.99 | 0.8633 |
| Topology terminology CCUF | 1.0 | 0.45 |

Table 10: Performance on non uniform filtration generation with different prompting strategies (Mean Rank ± std)

| Prompt Type | GPT 4o | Claude 3 7 |
|---|---|---|
| Few shot CoT | 52.09 ± 32.84 | 48.76 ± 29.99 |
| Rule emphasized | 53.11 ± 34.86 | 40.17 ± 31.31 |

Table 11: Post training on LLaMA-3.1-8B-Instruct with supervision on small graphs and evaluation on medium graphs.

| Prompt setting | 1D Simplex Counting | | Component Merge Time | | Component Count Under Filtration | |
|---|---|---|---|---|---|---|
| | Zero-shot | Post-train | Zero-shot | Post-train | Zero-shot | Post-train |
| Text style + Graph theoretic | 0.020 | 0.200 | 0.090 | 0.430 | 0.160 | 0.660 |
| Text style + Minimal | 0.000 | 0.160 | 0.040 | 0.290 | 0.090 | 0.400 |
| Text style + Topological | 0.000 | 0.040 | 0.080 | 0.430 | 0.330 | 0.480 |
| Matrix style + Topological | 0.000 | 0.060 | 0.140 | 0.250 | 0.210 | 0.480 |
| Code like + Topological | 0.000 | 0.260 | 0.010 | 0.370 | 0.220 | 0.540 |

Table 12: Summary of graph representation and task instruction styles. Full prompt examples are provided in the appendix.

| Prompt Style | Description |
|---|---|
| *Graph Representation Styles* | |
| Text-style format | Nodes and edges are described using plain natural language sentences, following human-like narration. |
| Code-like format | Graph elements are structured using syntax that resembles programming or configuration files, such as lists or dictionaries. |
| Matrix-based format | The graph is represented by a numerical adjacency matrix, requiring structural inference from tabular data. |
| *Task Instruction Styles* | |
| Topological formulation | Instructions are written using formal terminology from algebraic topology, such as simplex, homology, and filtration. |
| Graph-theoretic formulation | Tasks are described using standard graph terminology, avoiding domain-specific topological language. |
| Minimal formulation | Prompts are short and generic, with little or no explicit context, requiring the model to infer the task from minimal cues. |

