# OpenReview forum: "Large Language Models as Topological Thinkers: A Benchmark on Graph Persistent Homology"
_ICLR.cc/2026/Conference — Submitted to ICLR 2026_

### Official Review · Reviewer_tqDU · 2025-10-27

**Soundness:** 2
**Presentation:** 3
**Contribution:** 3
**Rating:** 4
**Confidence:** 3

**Summary:**

This paper introduces LLM4PH, the first benchmark designed to evaluate whether large language models (LLMs) can understand and reason about persistent homology (PH) — a core method in topological data analysis for capturing multi-scale graph structures. Results show that while LLMs handle low-level structural recognition (e.g., counting connected components) well, they struggle with higher-order reasoning such as dynamic filtration evolution and strategy design. The paper also explores prompt design, post-training effects, and integrated PH pipelines.

**Strengths:**

1. This paper is the first to test LLMs on topological reasoning, going beyond classical graph reasoning benchmarks.
2. This paper includes proprietary and open-source LLMs with multiple analysis dimensions

**Weaknesses:**

1. The PH reasoning tasks, while theoretically rich, may have narrow real-world applicability beyond benchmark testing.
2. Most experiments are on small graphs (≤30 nodes), raising scalability concerns.
3. Results show high sensitivity to prompt format and phrasing, questioning benchmark robustness.
4. The study reports accuracy metrics but offers limited qualitative insight into why models fail at topological abstraction.

**Questions:**

1. How do you ensure that improvements correspond to genuine topological reasoning rather than pattern memorization of graph-text associations?
2. How scalable is the benchmark to larger, real-world networks (e.g. thousands of nodes)?
3. Have you considered integrating explicit topological modules (e.g., PH libraries) into LLM reasoning loops to test hybrid neuro-symbolic approaches?

---

> ### Author Response · Authors · 2025-11-21
> **Thank you, Reviewer tqDU!**
>
> **W1:** The PH reasoning tasks, while theoretically rich, may have narrow real-world applicability beyond benchmark testing.
>
> **A1:** Thank you for the comment. While PH reasoning tasks are theoretically grounded, we believe they serve broader purposes than synthetic benchmarking alone. Persistent homology has growing practical relevance in areas such as neuroscience, computational biology, civil engineering, climate sciences, digital finance, and materials science, where users often need to interpret persistence diagrams, compare topological features, or select meaningful filtrations. To reflect this, our benchmark includes both synthetic and real-world graph datasets (e.g., BZR), and we evaluate models on tasks that simulate concrete components of PH workflows such as barcode comparison and filtration selection.
>
> Our aim is not to replicate full numerical pipelines, but to evaluate whether LLMs can engage with the symbolic and structural reasoning that underpins TDA practice.
>
> ---
> **W2:** Most experiments are on small graphs (≤30 nodes), raising scalability concerns.
> **Q2:** How scalable is the benchmark to larger, real-world networks (e.g. thousands of nodes)?
>
> **A2:**  Thank you for the suggestion. Due to the token limitations of current LLMs, our experiments are currently constrained to relatively small graphs. However, as discussed in **Page 9, Section 3.5 (“Large-Graph Scalability via Subgraph Sampling”)**, we also propose an experimental setting that targets larger datasets: we sample multiple subgraphs from larger graphs and track how model performance varies with the number of subgraph samples. This provides a practical approach to scaling PH reasoning under current model constraints.
>
> Finally, our benchmark framework supports the generation of larger synthetic graphs using the same task templates (e.g., for filtration, PD interpretation, and topological distance). We will clarify in the revision that scalability is a design goal, and our benchmark can flexibly support future large-graph evaluations as LLM context and reasoning capabilities improve.
>
> ---
> **W3:** Results show high sensitivity to prompt format and phrasing, questioning benchmark robustness.
>
> **A3:** Thank you for raising this important point. We acknowledge that LLMs are often sensitive to prompt formulation, which is why our benchmark explicitly incorporates multiple prompt formats and ablations to evaluate this effect. **Page 8, In Section 3.5, we include a Prompt Encoding and Task Formulation ablation** that tests models on both structured and naturalized prompt styles, as well as terminology variations. While performance does vary across formats, this variability itself is an intentional part of the benchmark design: it reveals how stable (or fragile) model reasoning is under realistic prompt changes.
>
> Rather than undermining robustness, we believe this sensitivity highlights a core motivation for the benchmark, probing not only whether LLMs can answer correctly, but also under what conditions their reasoning holds. We will clarify this aspect in the revision to distinguish between prompt fragility as a model limitation and as a diagnostic tool within our evaluation framework.

---

> > ### Author Response · Authors · 2025-11-21
> >
> > **Q1:** The study reports accuracy metrics but offers limited qualitative insight into why models fail at topological abstraction.
> >
> > **A1:** Thank you for this thoughtful comment. We agree that understanding *why* models fail is as important as *whether* they fail. In **Page 9, Section 3.5**, we include several qualitative analyses, for example, in the Compositional PH Pipeline task, we observe consistent failure modes where models correctly solve individual subtasks but cannot integrate them across steps. Similarly, in the PD Interpretation task, we find that models often hallucinate or miscount barcode features, indicating difficulty with symbolic tracking rather than just semantic parsing.
> >
> > We will expand this analysis in the revision by including more representative failure cases and explanations, illustrating how errors emerge from specific types of reasoning gaps (e.g., incorrect filtration logic, component merging, or diagram parsing). These qualitative insights complement the accuracy metrics and offer a clearer view into the topological abstraction limitations of current LLMs.
> >
> > ---
> > See **Q2** in **W2**
> >
> > ---
> > **Q3:** How do you ensure that improvements correspond to genuine topological reasoning rather than pattern memorization of graph-text associations?
> >
> > **A3:** To assess whether models rely on shallow pattern memorization of graph–text associations, we designed a controlled perturbation experiment. We tested models on three versions of the same tasks:
> >
> > - Group A: Edge lists are presented in sorted order (e.g., 0–1, 0–2, 1–3…).
> > - Group B: Edge lists are randomly shuffled (e.g., 11–5, 4–7, 8–2…).
> > - Group C: Based on Group A, but with small structural edits that change the topological property being queried (e.g., adding an edge to merge components or fill a cycle).
> >
> >   Results are shown below:
> >   |   | GPT-4o | Claude-3.7 |
> >   | :---- | :---- | :---- |
> >   | 0D Component Counting(A) | 0.4433   | 0.9833   |
> >   | 0D Component Counting(B) | 0.4133   | 0.9633   |
> >   | 0D Component Counting(C) | 0.36   | 0.93   |
> >   | 1D Simplex Counting(A) | 0.1533   | 0.2333   |
> >   | 1D Simplex Counting(B) | 0.1466   | 0.21   |
> >   | 1D Simplex Counting(C) | 0.09   | 0.24   |
> >
> >   We observe that models perform similarly in A and B, suggesting that performance is not heavily driven by edge order or memorized patterns. However, performance drops significantly in Group C, where subtle structural changes alter the correct answer. This indicates that model predictions are sensitive to actual topological content, not merely surface patterns. We will add this experiment to the appendix and clarify its implications in the main text.

---

> > > ### Author Response · Authors · 2025-11-21
> > >
> > > **Q4:** Have you considered integrating explicit topological modules (e.g., PH libraries) into LLM reasoning loops to test hybrid neuro-symbolic approaches?
> > >
> > > **A4** Thank you for the thoughtful suggestion. In addition to the hybrid pipelines reported in the main paper, we have further explored a more explicit neuro-symbolic integration by combining LLM-generated strategies with persistent homology (PH) libraries. Specifically, we designed an optimization experiment in which the LLM proposes and iteratively refines a filtration–selection function, while all topological computations (barcode generation and Wasserstein distance evaluation) are performed by PH tools.
> > >
> > > Experimental Setup:
> > >
> > > For each pair of real-world weighted graphs, the LLM is first asked to generate a scoring function over edge weights, which is then used to select a sequence of filtration values aimed at maximizing the Wasserstein distance between the two graphs. After computing the resulting distance via the PH library, we feed the current best function, its achieved distance, and the recent optimization history back to the LLM. The model then proposes a refined function. This loop continues until the distance fails to improve for three consecutive iterations or until a maximum of 20 iterations is reached. We conducted 20 runs each for GPT-4o and Claude.
> > >
> > > To guide reasoning, the prompt includes:
> > > - summary statistics and raw structures of the two graphs;
> > > - the last five iterations of weight distributions and achieved distances;
> > > - “expert features” describing structural roles of each weight value—MST importance (SMST), betweenness-based centrality (SCent), local degree-based relevance (SDeg), and statistical density (SDens).
> > > Each expert score is rank-normalized, and the LLM assigns weights to them to construct a combined scoring function. The top-10 weights are selected as the filtration values used by the PH library.
> > >
> > > **Results.**
> > > The table below summarizes optimization performance across 10 representative graph pairs:
> > >
> > > | PairID | Max Distance | First Max Iter | Total Iters |
> > > | ------ | ------------ | -------------- | ----------- |
> > > | 0      | 2.5001       | 5              | 10          |
> > > | 1      | 2.6512       | 1              | 6           |
> > > | 2      | 0.2871       | 4              | 9           |
> > > | 3      | 0.6921       | 1              | 6           |
> > > | 4      | 2.3407       | 1              | 6           |
> > > | 5      | 1.8001       | 1              | 6           |
> > > | 6      | 0.0228       | 5              | 10          |
> > > | 7      | 0.4069       | 2              | 7           |
> > > | 8      | 0.5899       | 1              | 6           |
> > > | 9      | 1.7708       | 2              | 7           |
> > >
> > > Across runs, the LLM-guided process typically converges within 6–10 iterations, and frequently identifies filtration strategies achieving substantially improved distances compared to initial proposals. These findings demonstrate that LLMs can interact meaningfully with symbolic PH modules: while they cannot reliably execute PH computations alone, they can generate and refine heuristic strategies that guide the numerical pipeline toward better topological separation.

---

> > > > ### Comment · Reviewer_tqDU · 2025-11-26
> > > >
> > > > Thanks for the extensive rebuttal. Some of my concerns have been well addressed. However, I still have some follow up questions.
> > > >
> > > > 1. I am curious about the ability of LLM on distinguishing non-isomorphism graph pairs. E.g. two graphs are non-isomorphic and can be distinguished by PH, but cannot be distinguished by WL-test [1]. This might be a little more challenging than the classification task proposed in the paper, but some researchers in the graph community might be interested in it.
> > > >
> > > > 2. For small scale graphs, enumerating the topological features like connected components and cycles can be more efficient than calling an LLM. There are also some GNN based works on substructure counting. What is the necessity of researching on the ability of LLM topological thinkers?
> > > >
> > > > 3. In the explanation about why models fail at topological abstraction, you mentioned that LLMs fail because they cannot combine the local information from each step. Have you considered that this is attributed to the way you use LLMs? Is it possible to solve this issue by designing some hierarchical structures or more advanced architectures like GraphRAG? Similarly, can the hallucinate issues be solved?
> > > >
> > > > [1] Horn, Max, et al. "Topological Graph Neural Networks." International Conference on Learning Representations.

---

> > > > > ### Author Response · Authors · 2025-11-27
> > > > > **Thanks very much for the feedback, Reviewer tqDU, we'll be back soon with the new experiments!**
> > > > >
> > > > > Thanks very much for the constructive feedback, Reviewer tqDU, we are working on the new experiments and we'll be back soon.

---

> > > > > ### Author Response · Authors · 2025-11-28
> > > > > **Answers for additional question.**
> > > > >
> > > > > **W1:** I am curious about the ability of LLM on distinguishing non-isomorphism graph pairs. E.g., two graphs are non-isomorphic and can be distinguished by PH, but cannot be distinguished by WL-test [1]. This might be a little more challenging than the classification task proposed in the paper, but some researchers in the graph community might be interested in it.
> > > > >
> > > > > **A1:** Thank you for the suggestion. To examine whether LLMs can distinguish non-isomorphic graphs that are indistinguishable by the WL-test, we conducted an additional experiment using the **CYCLES** dataset from Horn et al. (ICLR). These graphs share identical local neighborhoods (all nodes have degree 2), making them impossible for WL to distinguish, while PH assigns different Betti numbers.
> > > > >
> > > > > Dataset statistics (200 samples):
> > > > > | Betti (β₀, β₁) | Count | Percentage |
> > > > > |----------------|--------|------------|
> > > > > | (1, 1) | 115 | 57.5% |
> > > > > | (3, 3) | 18 | 9.0% |
> > > > > | (4, 4) | 30 | 15.0% |
> > > > > | (5, 5) | 15 | 7.5% |
> > > > > | (6, 6) | 22 | 11.0% |
> > > > >
> > > > > Experimental Setup:
> > > > >
> > > > > We evaluated GPT-4o and Claude-3-7 on two PH-based tasks:
> > > > > - **0D Component Counting**
> > > > > - **1D Simplex Counting**
> > > > > After obtaining the results for Betti numbers, we further conducted a **non-isomorphic graph distinction task** using the computed topological features.
> > > > >
> > > > > Results:
> > > > > | Task | GPT-4o | Claude-3-7 |
> > > > > |------|--------|------------|
> > > > > | 0D Component Counting | 0.94 | 0.92 |
> > > > > | 1D Simplex Counting | 0.775 | 0.985 |
> > > > > | Non-isomorphic Graph Distinction | 0.9800(TP: 81, TN: 115, FP: 0, FN: 4) | 1  |
> > > > >
> > > > > Findings:
> > > > >
> > > > > Both models achieve high accuracy on distinguishing these WL-indistinguishable graphs, especially on 0D component counting and 1D simplex counting. This indicates that LLMs are **not constrained by WL-type local aggregation** and can exploit **global structural cues** provided in the textual graph representation. These results suggest the interesting and arguably unexpected phenomenon that LLM attention mechanisms provide a global receptive field that enables distinguishing certain non-isomorphic graph pairs beyond the capacity of WL.
> > > > > We will include this experiment in the revised manuscript, as it reinforces the claim that LLMs exhibit structural reasoning capabilities that differ from GNN/WL-style models, and connects our benchmark with broader interests in the graph learning community.
> > > > >
> > > > > **W2:** For small scale graphs, enumerating the topological features like connected components and cycles can be more efficient than calling an LLM. There are also some GNN-based works on substructure counting. What is the necessity of researching the ability of LLM topological thinkers?
> > > > >
> > > > > **A2:** Thank you for the question. We agree that LLMs are in {\bf no way} intended to replace classical PH algorithms or GNN-based substructure counters, especially on small graphs where exact computation is trivial. The goal of our benchmark is fundamentally different: rather than using LLMs as topological *computers*, we aim to evaluate whether LLMs can act as **topological reasoning assistants**. Furthermore, we hypothesise that at some point such LLM topological reasoning assistants can be used to help in sequential updates of PH features in various learning tasks involving dynamic graphs.
> > > > >
> > > > > It is important to note that in various real TDA workflows, many tasks require symbolic reasoning, qualitative interpretation, and decision-making beyond numerical computations, such as choosing appropriate filtration strategies, interpreting persistence diagrams, explaining topological behavior, or integrating PH outputs with domain knowledge. These tasks cannot be solved by PH libraries or GNN models alone.
> > > > >
> > > > > Our results in Section 3.5 further show that LLMs struggle even on conceptually simple PH operations, which highlights important limitations in their ability to connect graph structure with topological abstraction. Understanding these limitations is essential for developing future neuro-symbolic systems that combine LLM reasoning with classical PH tools.
> > > > >
> > > > > In summary, the objective is not to outperform existing PH algorithms, but to study whether current LLMs possess the symbolic, compositional, and interpretive capabilities necessary to support TDA in practice, and what knowledge gaps exist right now to guide the LLM improvement in these directions.

---

> > > > > ### Author Response · Authors · 2025-11-28
> > > > >
> > > > > **W3:** In the explanation about why models fail at topological abstraction, you mentioned that LLMs fail because they cannot combine the local information from each step. Have you considered that this is attributed to the way you use LLMs? Is it possible to solve this issue by designing some hierarchical structures or more advanced architectures like GraphRAG? Similarly, can the hallucination issues be solved?
> > > > >
> > > > > **A3:** Thank you for raising this point. Our error analysis shows that two major sources of failure are **fact forgetting** and **rule forgetting**, both of which arise when long-range consistency must be maintained across multiple filtration steps. To examine whether more hierarchical or state-aware prompting can mitigate these issues, as the reviewer suggests, we designed an additional experiment incorporating an explicit **memory mechanism** inspired by agent-style LLM research.
> > > > >
> > > > > Experimental Setup:
> > > > > At each filtration value (edges sorted from low to high weight), the LLM receives:
> > > > >
> > > > > 1. Newly activated edges at this step
> > > > > 2. The **state memory** produced by the LLM in the previous step
> > > > > 3. The corresponding PH update rule (component merging or cycle creation)
> > > > >
> > > > > Two tasks were tested (100 samples, medium-difficulty tasks):
> > > > > - **Component Merge Time (0D):**
> > > > >   The LLM updates a set of connected components and determines whether a merge occurs.
> > > > > - **Simplex Birth Time (1D):**
> > > > >   The LLM updates the active-edge list and determines whether a new 1-cycle is formed.
> > > > >
> > > > > Results:
> > > > > | Task | GPT-4o | Claude-3-7 |
> > > > > |------|--------|------------|
> > > > > | 0D merge (with memory) | 0.70 | 0.975 |
> > > > > | 0D merge (single-shot) | 0.50 | 0.275 |
> > > > > | 1D birth (with memory) | 0.35 | 0.475 |
> > > > > | 1D birth (single-shot) | 0.275 | 0.375 |
> > > > >
> > > > > Findings:
> > > > > The memory-based hierarchical setup significantly improves performance demonstrating that explicit state management can partially alleviate information-loss issues. However, the improvement is not uniform across tasks, and even with memory, models still struggle with multi-step topological updates (e.g., 1D cycle creation).

---

### Official Review · Reviewer_gB2w · 2025-10-29

**Soundness:** 3
**Presentation:** 3
**Contribution:** 2
**Rating:** 4
**Confidence:** 4

**Summary:**

The paper proposes a dataset of graphs and a set of tasks to asses the capacity of pretrained LLMs to understand topological tasks, such as predicting the connected components and subsets of persistent homology. The tasks consist of predicting connected components and birth and death times of cycles as well as predicting the critical edges / nodes for them. Harder tasks consist of finding the  best filter function to separate graphs.
A core part  of the paper lies in the evaluation of pretrained models on these task through either their API (for the proprietary models) or on a cluster when the model is open source.

**Strengths:**

Overall there is some merit in the idea that it can be interesting to understand how well an LLM can understand harder tasks such as predicting filter functions and computing higher order homology. The paper is well-written and the code is provided and well documented.

**Weaknesses:**

Overall the question why the predication of graphs could be potentially interesting is somewhat hard to understand. Hence the motivation and potential impact on either the field of TDA or machine learning seems somewhat hard to understand. In particular, fundamentally LLMs are designed to predict text rather than topological features and therefore evaluating them on this specific task would need more motivation.
The dataset provided consists of graphs, but the motivation to use exactly this set of graphs is a little bit unclear to me.

**Questions:**

In line with the weaknesses, there are some answers to be provided for a broader impact on the community.
- What would the reader, or user, compel to use their dataset, as compared to an arbitrary graph dataset? This question is a little unclear to the reviewer.
- What would be the broader impact of this work for either the TDA community or LLMs, if we would want to train a model for a particular problem, predicting it with an LLM would not be feasible anyway and also not expected and we would want to do this at runtime / training time as well.

---

> ### Author Response · Authors · 2025-11-21
> **Thank you, Reviewer gB2w!**
>
> **W1:** Overall the question why the predication of graphs could be potentially interesting is somewhat hard to understand.
>
> **A1:** Thank you for raising this important point. Our motivation comes from two complementary perspectives. **In page 2 you can find the paragraph that we have revised.**
>
> First, from the mathematical reasoning perspective, recent benchmarks such as MathOdyssey[1] and HighMATH[2] show that evaluating LLMs on concrete mathematical pipelines is essential for understanding their non linguistic reasoning abilities. Persistent homology provides a structured sequence of operations including filtrations and birth death computations. Our benchmark follows this workflow and tests whether LLMs can reconstruct these operations from language alone. Section 3.5 further evaluates whether models can combine them into a complete PH pipeline.
>
> Second, from the graph understanding perspective, recent work shows that LLMs can partially recognise high order graph patterns[3] and even hypergraph structures[4] from textual descriptions. Persistent homology naturally captures such higher order properties. Our tasks therefore examine whether LLMs can infer global topological features of graphs rather than rely on surface level edge descriptions. The prompt encoding ablations in Section 3.5 explicitly test this.
>
> [1] Fang M, Wan X, Lu F, et al. Mathodyssey: Benchmarking mathematical problem-solving skills in large language models using odyssey math data[J]. Scientific Data
> [2] Liu Y, Zhang M, Xiong B, et al. HighMATH: Evaluating Math Reasoning of Large Language Models in Breadth and Depth[C]//The 2025 Conference on Empirical Methods in Natural Language Processing.
> [3] Dai X, Qu H, Shen Y, et al. How Do Large Language Models Understand Graph Patterns? A Benchmark for Graph Pattern Comprehension[C]//The Thirteenth International Conference on Learning Representations.
> [4] Feng Y, Yang C, Hou X, et al. Beyond Graphs: Can Large Language Models Comprehend Hypergraphs?[C]//The Thirteenth International Conference on Learning Representations.
>
> ---
> **W2:** What would the reader, or user, compel to use their dataset, as compared to an arbitrary graph dataset? This question is a little unclear to the reviewer.
>
> **A2:** Thank you for the question. The key reason to use the datasets provided in our benchmark rather than an arbitrary graph collection is that our tasks require controlled and balanced topological structures. For example, in the Component Counting task, we need exactly balanced sets of graphs with one, two, three and four components. This level of structural control cannot be guaranteed by arbitrary graph datasets and is necessary to ensure fair and interpretable evaluation.
>
> In addition, our benchmark includes ten subtasks and multiple synthetic graph generators designed to produce specific filtrations and persistent homology patterns. These controlled constructions allow users to test LLM behaviour under well defined topological conditions. We provide the full data generation pipeline in the codebase, so users may either use our datasets directly or generate their own variants using the same methodology.
>
> ---
> **W3:** What would be the broader impact of this work for either the TDA community or LLMs, if we would want to train a model for a particular problem, predicting it with an LLM would not be feasible anyway and also not expected and we would want to do this at runtime / training time as well.
>
> **A3:** Thank you for raising this point. Our goal is not to propose LLMs as replacements for persistent homology algorithms, nor to suggest that PH should be computed by LLMs at runtime. Instead, the broader impact of this work lies in providing a controlled benchmark for understanding the limits of current LLM reasoning when faced with non-linguistic, algorithmic, and topological structure.
>
> For the TDA community, our benchmark aims to clarify the potential of LLMs as assistants for topological data analysis. It is well known that LLMs can provide high-level conceptual explanations, but it remains unclear whether they can support more substantial tasks. Through this benchmark, we hope to examine whether LLMs can meaningfully assist with components such as selecting filtration functions, qualitatively analyzing large numbers of persistence diagrams, and other practical steps that arise in TDA workflows.
>
> For the LLM community, our benchmark isolates a class of reasoning problems that require algorithmic structure rather than pattern memorisation. This complements existing mathematical benchmarks and highlights the gap between surface-level textual reasoning and genuine structural abstraction. Such insights can guide the development of models that incorporate stronger geometric or algorithmic priors.

---

> > ### Comment · Reviewer_gB2w · 2025-11-24
> >
> > Thank you very much for the extensive answers. After reading the other reviews I'm still hesitant regarding the applicability of the method and decided keep my score as is.

---

> > > ### Author Response · Authors · 2025-11-27
> > > **Thanks very much for the feedback, Reviewer gB2w! We'll try to showcase some examples to address your utility questions.**
> > >
> > > Thanks very much for the feedback, Reviewer gB2w! We'll try to present some additional examples to address your question on applicability of our results.

---

### Official Review · Reviewer_XKxG · 2025-10-30

**Soundness:** 3
**Presentation:** 3
**Contribution:** 2
**Rating:** 4
**Confidence:** 4

**Summary:**

The paper introduces a benchmark designed to evaluate the ability of LLMs to reason about graph topology through the lens of persistent homology.  The benchmark comprises a hierarchically organised set of tasks and is accompanied by an in-depth comparative study of the performance of  multiple LLMs, as well as analyses of prompt formulation, graph representation, and recurring error types.

**Strengths:**

[S1] The paper is well written and generally easy to follow.

[S2] The benchmark is thoughtfully designed and covers a good range of PH-related reasoning tasks.

[S3] I particularly appreciate the error analysis presented in Section 3.4, which provides a clear window into model behaviour and failure modes.

**Weaknesses:**

[W1] My main concern lies in the motivation for this work. Persistent homology is a rather specialised area, and it is unclear why developing LLM benchmarks for PH reasoning is of broad importance. While the experiments are interesting, the contribution may be more suitable for a more focused venue (e.g., a workshop or conference on graphs and topology in AI) unless the authors can articulate a clearer connection to the general capabilities or limitations of LLMs or better justify the importance for LLMs to be able to perform these TDA computations.

[W2] The three “simple” tasks defined in the paper do not strongly reflect persistent homology reasoning—they could just as well appear in a general graph reasoning benchmark. It would strengthen the work to include additional tasks that more explicitly capture PH-specific concepts within this task family.

[W3] In Section 3.1, the finding that LLMs perform worse when PH-specific terminology is used seems unsurprising. This likely reflects the models’ limited familiarity with the jargon of a niche field rather than any deeper issue with topological reasoning itself. Similarly, the statement that “LLMs are capable of expressing meaningful topological reasoning only when guided by structured prompts and constrained options” may stem more from unfamiliar terminology than from fundamental reasoning limitations. If so, this insight is less significant than it initially appears.

**Questions:**

[Q1] Could you further justify why PH reasoning is an important or representative capability for evaluating LLMs?

[Q2] Could you expand on the role of terminology in your results? You briefly mention this in Section 3.5, but it’s unclear whether your main experiments rely on PH-specific language. Why is it important, or interesting, for an LLM to understand this terminology rather than simply solving the task when phrased in more neutral terms? Couldn’t this familiarity be induced through an appropriate system prompt?

---

> ### Author Response · Authors · 2025-11-21
> **Thanks a lot, Reviewer XKxG!**
>
> **W1:** My main concern lies in the motivation for this work. Persistent homology is a rather specialised area, and it is unclear why developing LLM benchmarks for PH reasoning is of broad importance. While the experiments are interesting, the contribution may be more suitable for a more focused venue (e.g., a workshop or conference on graphs and topology in AI) unless the authors can articulate a clearer connection to the general capabilities or limitations of LLMs or better justify the importance for LLMs to be able to perform these TDA computations.
>
> **A1:** Thank you for the thoughtful question. **In page 2 you can find the paragraph that we have revised.** Our benchmark is motivated by the compositional structure of the persistent homology (PH) pipeline. PH inherently combines both mathematical reasoning and graph structural understanding.  In particular, PH is both a well defined mathematical pipeline (with its own formalism) and an increasingly popular machinery for capturing higher-order graph structures in ML, which connects our work to recent mathematical reasoning benchmarks \[1,2\] and to emerging evaluations of high-order graph understanding \[3,4\].
>
> From an applied standpoint, we aim to understand whether LLMs can function as reliable assistants for PH workflows. Section 3.5 shows that performance drops sharply when subtasks are composed into an end-to-end PH pipeline, indicating persistent difficulties with multi-step algorithmic reasoning. This provides insight into broader limitations of current LLMs rather than focusing narrowly on PH.
>
> Finally, several of our tasks require models to infer global topological features that are not explicitly stated in the edge list. This aligns with recent findings that LLMs struggle to generalise beyond local or semantic graph patterns. Our results reinforce this gap through a different modality: multi-scale topological structure.
>
> In summary, although PH may appear somewhat specialised, our benchmark uses it as a rigorous testbed for general LLM reasoning, highlighting limitations in abstraction, compositionality and structural understanding. At the same time, our benchmark also paves a way to adoption of LLM reasoning in higher-order graph learning \[5\], including but not limited to topological deep learning. We will make this motivation clearer in the revision.
>
> \[1\] Fang M, Wan X, Lu F, et al. Mathodyssey: Benchmarking mathematical problem-solving skills in large language models using odyssey math data\[J\]. Scientific Data, 2025, 12(1): 1392\.
> \[2\] Liu Y, Zhang M, Xiong B, et al. HighMATH: Evaluating Math Reasoning of Large Language Models in Breadth and Depth\[C\]//The 2025 Conference on Empirical Methods in Natural Language Processing. https://openreview. net/forum. 2025\.
> \[3\] Dai X, Qu H, Shen Y, et al. How Do Large Language Models Understand Graph Patterns? A Benchmark for Graph Pattern Comprehension\[C\]//The Thirteenth International Conference on Learning Representations.
> \[4\] Feng Y, Yang C, Hou X, et al. Beyond Graphs: Can Large Language Models Comprehend Hypergraphs?\[C\]//The Thirteenth International Conference on Learning Representations.
> \[5\]Papamarkou T. et al. Position: Topological Deep Learning is the New Frontier for Relational Learning, ICML2024

---

> > ### Author Response · Authors · 2025-11-21
> >
> > **W2:** The three “simple” tasks defined in the paper do not strongly reflect persistent homology reasoning—they could just as well appear in a general graph reasoning benchmark. It would strengthen the work to include additional tasks that more explicitly capture PH-specific concepts within this task family.
> >
> > **A2:** Thank you for the suggestion. Following a similar recommendation from Reviewer 9E47, we have added two additional tasks that are more directly tied to persistent homology (PH) reasoning. These include:(1) Persistence Diagram Interpretation task and (2) Topological Distance Reasoning task. **In page 3 you can find the paragraph that we have added.**
> >
> > (1) Persistence Diagram Interpretation
> >
> > Objective: Evaluate whether LLMs can correctly parse a PD represented in text and perform basic logical operations.
> >
> > Data Generation:
> > - 300 synthetic PDs generated directly as barcodes (no geometric computation)
> > - All numerical values sampled from integers 1–10
> > - Each PD contains both H0 and H1 intervals (10–15 bars total)
> > - A random filtration value t ∈ {1, …, 10} is selected per instance
> >
> > Tasks:
> > - Task A (Static Reading): Compute the longest finite lifetime in H0 (death minus birth; ignoring infinite bars).
> > - Task B (Filtered Reading): Count the number of H1 features alive at time t (excluding features that die exactly at t).
> >
> > Evaluation Metric: Accuracy.
> >
> > Results:
> >
> > | Task | GPT-4.1-mini | Claude-3-7 |
> > |------|---|-----|
> > | max_lifetime| 0.6366 | 0.8766|
> > | filter_count| 0.4833|0.2433|
> >
> > These results show that even when persistent diagrams are provided directly in textual form, model performance remains far from reliable. PD interpretation alone already represents a nontrivial reasoning challenge for current LLMs.
> >
> > (2) Topological Distance Reasoning
> >
> > Objective: Test whether LLMs can infer topological similarity between diagrams by reasoning over Wasserstein distances.
> >
> > Data Generation:
> > - 300 test groups
> > - Each group contains 4 independently generated PDs (A, B, C, D)
> > - PD generation rules identical to the previous subtask
> >
> > Ground Truth Computation:
> > - Combine H0 and H1 bars for each PD
> > - Compute all 6 pairwise Wasserstein distances among the four PDs
> > - The pair with the smallest distance is used as the correct answer
> >
> > Task:
> > Given four PDs in text format, the model is asked:  "Which two diagrams are topologically most similar?" Expected output: a pair such as `[A, C]`.
> >
> > Evaluation Metric: Accuracy.
> >
> > Results:
> > | Model| Accuracy |
> > |--|--|
> > | GPT-4.1-mini | 0.0266|
> > | Claude-3-7|0.18|
> >
> > These results indicate that LLMs struggle significantly with PH-specific distance reasoning, even when diagrams are explicitly provided. This confirms that PH introduces a fundamentally different and more demanding form of structural abstraction compared to traditional math or graph benchmarks.

---

> > > ### Author Response · Authors · 2025-11-21
> > >
> > > **W3:** In Section 3.1, the finding that LLMs perform worse when PH-specific terminology is used seems unsurprising. This likely reflects the models’ limited familiarity with the jargon of a niche field rather than any deeper issue with topological reasoning itself. Similarly, the statement that “LLMs are capable of expressing meaningful topological reasoning only when guided by structured prompts and constrained options” may stem more from unfamiliar terminology than from fundamental reasoning limitations. If so, this insight is less significant than it initially appears.
> > >
> > > **A3:** Thank you for this insightful observation. To assess whether the use of PH-specific terminology significantly affects model performance, we conducted a terminology ablation study across three medium-difficulty tasks, comparing topological phrasing (e.g., “H0 class”) with graph-theoretic alternatives (e.g., “connected components”) while keeping the underlying operations identical. In page 8 you can find the paragraph that we have added.
> > >
> > > Objective: Evaluate the sensitivity of LLMs to domain terminology by testing whether replacing PH terms (e.g., “H0 class”) with graph-theoretic terms (e.g., “connected components”) affects task accuracy.
> > >
> > > Experimental Design:
> > > \- Three medium-difficulty tasks were selected.
> > > \- For each task, we created two prompt versions:
> > >   \- Graph-theory wording
> > >   \- Topology wording
> > > \- Only terminology differs; the underlying operations remain the same.
> > > \- 300 test instances per condition.
> > > \- Models evaluated: GPT-4.1-mini and GPT-4o.
> > > \- Metric: Accuracy.
> > >
> > > Example Prompt Pair:
> > >
> > > (1) Graph-theory wording
> > > Task: Count how many **graph components** appear when **the threshold is 3**.
> > > You can follow these steps:
> > > Step1: Collect all edges whose weight is less than or equal to 3\.
> > > Step2: Construct a graph from these edges.
> > > Step3: Count the number of **connected components in this graph**.
> > > Rule: A vertex appears in the **structure** only when one of its incident edges is included.
> > >
> > > (2) Topology wording
> > > Task: Count how many **H0 classes** exist at **filtration value 3**.
> > > You can follow these steps:
> > > Step1: Select all edges with filtration value (weight) less than or equal to 3\.
> > > Step2: Build the 1-skeleton of the filtered complex using these edges.
> > > Step3: Count the number of 0**\-dimensional homology classes (H0) at this filtration stage.**
> > > Rule: A vertex enters the **complex** only when at least one adjacent edge is included.
> > >
> > > Results:
> > >
> > > | Condition                  | GPT-4.1-mini | GPT-4o   |
> > > |---------------------------|--------------|----------|
> > > | Graph terminology – birth | 0.67         | 0.33     |
> > > | Topology terminology – birth | 0.6533    | 0.3233   |
> > > | Graph terminology – merge | 0.02         | 0.5533   |
> > > | Topology terminology – merge | 0.0066    | 0.57     |
> > > | Graph terminology – CCUF  | 0.99         | 0.8633   |
> > > | Topology terminology – CCUF | 1.0        | 0.45     |
> > >
> > > The results show task-dependent effects. In the “birth” task, both GPT-4o and GPT-4.1-mini perform nearly identically under both phrasings, suggesting terminology has minimal impact in that setting. However, in the “CCUF” task, GPT-4o shows a noticeable drop (from 0.8633 to 0.45) when using PH-specific language, indicating that terminology may introduce added difficulty in tasks involving multi-step compositional reasoning.
> > >
> > > These findings suggest that terminology can influence performance in certain contexts, but it does not uniformly account for failure modes across all tasks. The overall gap between model outputs and correct PH logic cannot be fully attributed to unfamiliar jargon alone.

---

> > > > ### Author Response · Authors · 2025-11-21
> > > >
> > > > **W4:** Could you further justify why PH reasoning is an important or representative capability for evaluating LLMs?
> > > >
> > > > **A4:**
> > > >
> > > > Thank you for the question. Persistent homology (PH) reasoning aligns with the goals of existing LLM benchmarks but introduces evaluation aspects that have not been covered before. Similar to mathematical benchmarks such as MATH, MathOdyssey and HighMATH, our tasks require models to follow a mathematically defined procedure. Similar to graph benchmarks, they require interpreting structural information from textual descriptions of edges. In this sense, PH reasoning fits naturally within the landscape of structured reasoning benchmarks.
> > > >
> > > > The novelty comes from the fact that PH integrates these two directions into a single, multi-step pipeline that no previous benchmark targets. Unlike symbolic math tasks, PH requires models to reconstruct geometric and topological behaviour across scales, including filtrations, birth and death events and barcode-level comparisons. Unlike typical graph tasks, the required reasoning is not local or combinatorial but global and multi-scale. This combination of geometric abstraction, algorithmic compositionality and higher-order structure inference is unique to PH and reveals reasoning failures that do not surface in existing benchmarks.
> > > >
> > > > Our empirical study shows exactly this: while models can occasionally solve isolated steps, performance drops sharply when subtasks must be composed into an end-to-end PH workflow. This indicates that PH reasoning probes a general limitation of LLMs in structured and compositional reasoning, rather than a narrow domain-specific skill. We will clarify this benchmark positioning more explicitly in the revision.
> > > >
> > > > **W5:** Could you expand on the role of terminology in your results? You briefly mention this in Section 3.5, but it’s unclear whether your main experiments rely on PH-specific language. Why is it important, or interesting, for an LLM to understand this terminology rather than simply solving the task when phrased in more neutral terms? Couldn’t this familiarity be induced through an appropriate system prompt?
> > > >
> > > > **A5:** Thank you for the thoughtful question. We agree that distinguishing between terminology sensitivity and structural reasoning capability is critical. In response, we have conducted a terminology ablation study (**Page 8, Section 3.5**), rewriting prompts in both topological and graph-theoretic language while preserving task logic. While accuracy remained nearly identical for simpler tasks (e.g., “birth”), performance degraded notably for more complex ones (e.g., “CCUF”) when PH-specific language was used. This suggests that terminology affects performance, particularly in tasks involving multi-stage reasoning.
> > > >
> > > > That said, we argue that understanding PH terminology is not merely cosmetic. Our motivation is to assess whether LLMs can act as effective assistants in practical TDA workflows, where concepts like “H0 class,” “filtration,” or “1-skeleton” are used explicitly and require domain-accurate parsing. In these scenarios, users are unlikely to rephrase tasks into simplified terms, and success depends on the model’s ability to parse and apply domain-specific language.
> > > >
> > > > While it is possible that certain terminology could be introduced via a carefully designed system prompt, our findings suggest that such superficial guidance does not consistently induce correct multi-step reasoning. The terminology is often embedded within logical operators or structural references (e.g., “birth–death pairing”), and mere substitution does not equip the model to manipulate the concepts effectively.

---

> ### Comment · Reviewer_XKxG · 2025-11-27
>
> I thank the authors for their thoughtful and detailed response, which has addressed most of my concerns. I will be raising my score accordingly.

---

> > ### Author Response · Authors · 2025-11-27
> > **Thanks very much, Reviewer XKxG!**
> >
> > Dear Reviewer XKxG,
> >
> > Thanks very much for appreciating the novelty and contribution of our paper and, of course, for raising the score! If you think that you could champion the paper,  we'd be supergrateful.
> >
> > We'll be meanwhile running new experiments to answer additional questions from reviewers and will keep updating the paper.
> >
> > Authors of the paper.

---

### Official Review · Reviewer_9E47 · 2025-11-02

**Soundness:** 4
**Presentation:** 3
**Contribution:** 4
**Rating:** 6
**Confidence:** 5

**Summary:**

This paper proposes LLM4PH, the first benchmark for evaluating large language models (LLMs)’ ability to understand and apply persistent homology (PH) on graphs. It decomposes the PH pipeline into 4 difficulty levels (10 subtasks), uses 3 sizes of synthetic graphs and 3 real molecular graph datasets (BZR, COX2, LDHFR), and assesses 9 LLMs. The work aims to bridge LLMs’ discrete graph reasoning and continuous topological abstraction, providing insights into structure-aware scientific reasoning.

**Strengths:**

1. Originality: Fills a gap by being the first benchmark targeting LLMs’ PH understanding, addressing the lack of topological reasoning evaluation in existing graph benchmarks.
2. Rigorous Experiments: Evaluates diverse LLMs (proprietary/open-source) with controlled graph sizes, and conducts cross-task ablations (prompt encoding, post-training) for in-depth analysis.
3. Clarity: Well-structured with clear task definitions, result tables, and appendices for reproducibility.
4. Significance: Addresses LLMs’ limitations in non-Euclidean topological reasoning, with implications for fields like molecular biology and social network analysis.

**Weaknesses:**

1. Lacks independent subtasks for "persistence diagram interpretation" and "topological distance reasoning"—core PH steps.
No Terminology Ablation: Fails to verify if low accuracy in "1D Simplex Counting" stems from "1-simplex" terminology (vs. "edge") via ablation.
2. Does not test prompt engineering (e.g., CoT) to fix errors like "context loss" in non-uniform filtration generation.
3. No Traditional PH Tool Comparison: Omits performance contrasts between LLMs, full traditional PH pipelines, and hybrid pipelines.
4. Lacks key works on simplicial complex identification (e.g., Hypergraph Isomorphism Computation, Reinterpreting Hypergraph Kernels: Insights Through Homomorphism Analysis).

**Questions:**

see weaknesses

---

> ### Author Response · Authors · 2025-11-21
> **Thank you very much, Reviewer 9E47!**
>
> **W1:** Lacks independent subtasks and No Terminology Ablation.
>
> **A1.1:** Added Subtasks.
>
> Thank you for pointing out this limitation. We have added two independent subtasks that directly target core components of persistent homology. **In page 3 you can find the paragraph that we have added.**
>
> (1) Persistence Diagram Interpretation
>
> Objective: Evaluate whether LLMs can correctly parse a PD represented in text and perform basic logical operations.
>
> Data Generation:
> - 300 synthetic PDs generated directly as barcodes (no geometric computation)
> - All numerical values sampled from integers 1–10
> - Each PD contains both H0 and H1 intervals (10–15 bars total)
> - A random filtration value t ∈ {1, …, 10} is selected per instance
>
> Tasks:
> - Task A (Static Reading): Compute the longest finite lifetime in H0 (death minus birth; ignoring infinite bars).
> - Task B (Filtered Reading): Count the number of H1 features alive at time t (excluding features that die exactly at t).
>
> Evaluation Metric: Accuracy.
>
> Results:
>
> | Task | GPT-4.1-mini | Claude-3-7 |
> |------|---|-----|
> | max_lifetime| 0.6366 | 0.8766|
> | filter_count| 0.4833|0.2433|
>
> These results show that even when persistent diagrams are provided directly in textual form, model performance remains far from reliable. PD interpretation alone already represents a nontrivial reasoning challenge for current LLMs.
>
> (2) Topological Distance Reasoning
>
> Objective: Test whether LLMs can infer topological similarity between diagrams by reasoning over Wasserstein distances.
>
> Data Generation:
> - 300 test groups
> - Each group contains 4 independently generated PDs (A, B, C, D)
> - PD generation rules identical to the previous subtask
>
> Ground Truth Computation:
> - Combine H0 and H1 bars for each PD
> - Compute all 6 pairwise Wasserstein distances among the four PDs
> - The pair with the smallest distance is used as the correct answer
>
> Task:
> Given four PDs in text format, the model is asked:  "Which two diagrams are topologically most similar?" Expected output: a pair such as `[A, C]`.
>
> Evaluation Metric: Accuracy.
>
> Results:
> | Model| Accuracy |
> |--|--|
> | GPT-4.1-mini | 0.0266|
> | Claude-3-7|0.18|
>
> These results indicate that LLMs struggle significantly with PH-specific distance reasoning, even when diagrams are explicitly provided. This confirms that PH introduces a fundamentally different and more demanding form of structural abstraction compared to traditional math or graph benchmarks.
>
> **A1.2:** Terminology Ablation.
>
> To evaluate whether model performance is influenced by PH-specific terminology rather than by the underlying reasoning steps, we conducted a terminology ablation experiment. This experiment compares LLM performance under two prompt styles: graph-theoretic vocabulary and topological vocabulary, while keeping the task semantics identical. **In page 3 you can find the paragraph that we have added. **
>
> Objective: Evaluate the sensitivity of LLMs to domain terminology by testing whether replacing PH terms (e.g., “H0 class”) with graph-theoretic terms (e.g., “connected components”) affects task accuracy.
>
> Experimental Design:
> - Three medium-difficulty tasks were selected.
> - For each task, we created two prompt versions:
>   - Graph-theory wording
>   - Topology wording
> - Only terminology differs; the underlying operations remain the same.
> - 300 test instances per condition.
> - Models evaluated: GPT-4.1-mini and GPT-4o.
> - Metric: Accuracy.
>
> Example Prompt Pair:
>
> (1) Graph-theory wording
> Task: Count how many **graph components** appear when **the threshold is 3**.
> You can follow these steps:
> Step1: Collect all edges whose weight is less than or equal to 3\.
> Step2: Construct a graph from these edges.
> Step3: Count the number of **connected components in this graph**.
> Rule: A vertex appears in the **structure** only when one of its incident edges is included.
>
> (2) Topology wording
> Task: Count how many **H0 classes** exist at **filtration value 3**.
> You can follow these steps:
> Step1: Select all edges with filtration value (weight) less than or equal to 3\.
> Step2: Build the 1-skeleton of the filtered complex using these edges.
> Step3: Count the number of 0**\-dimensional homology classes (H0) at this filtration stage.**
> Rule: A vertex enters the **complex** only when at least one adjacent edge is included.
>
> Results:
>
> | Condition| GPT-4.1-mini | GPT-4o   |
> |--|--|-|
> | Graph – birth | 0.67| 0.33     |
> | Topology – birth | 0.6533 0.3233   |
> | Graph – merge | 0.02| 0.5533   |
> | Topology – merge | 0.0066| 0.57     |
> | Graph – CCUF  | 0.99| 0.8633   |
> | Topology – CCUF | 1.0| 0.45     |
>
> Performance differences between graph-theoretic and PH-specific phrasings are small and inconsistent across tasks. This demonstrates that:
> 1. Low accuracy is not driven by unfamiliar PH terminology, and
> 2. The core difficulty lies in the structural and algorithmic reasoning required by PH tasks, rather than vocabulary matching.

---

> > ### Author Response · Authors · 2025-11-21
> >
> > **W2:** Does not test prompt engineering (e.g., CoT) to fix errors like "context loss" in non-uniform filtration generation.
> >
> > **A2:** Thank you for raising this concern. To evaluate whether prompt engineering techniques can mitigate errors such as “context loss” in non-uniform filtration generation, we conducted a prompt-ablation experiment. This experiment tests whether improved prompting truly enhances performance on this structurally challenging PH task. **In page 8 you can find the paragraph that we have added.**
> >
> > Objective: Test whether Few-shot Chain-of-Thought (CoT) or rule-emphasized prompting improves model performance in non-uniform filtration generation.
> >
> > Experimental Design:
> >
> > - Task: non-uniform filtration generation.
> > - Two improved prompt variants:
> >   - Few-shot Chain-of-Thought (CoT)
> >   - Rule-emphasized prompting
> > - 300 test instances for each prompting strategy.
> > - Models: GPT-4o and Claude.
> > - Metric: Mean Rank (lower is better).
> >
> > Prompt Variant 1: Few-shot Chain-of-Thought (CoT)
> >
> > - Provides a full worked example showing the complete reasoning process for a small graph.
> > - Teaches the model by demonstration, encouraging it to imitate a step-by-step PH analysis.
> > - Emphasizes example-driven reasoning rather than strict rule enforcement.
> >
> > Prompt Variant 2: Rule-Emphasized Prompting
> > - Does not give an example; instead, it supplies explicit, strict rules the model must follow.
> > - Clearly defines edge activation logic, H0/H1 updates and mandatory verification steps.
> > - Forces the model to reason through a constrained, stepwise procedure.
> >
> > Results:
> >   | Prompt Type| GPT-4o         | Claude        |
> >   |-|--|---------------|
> >   | Few-shot Mean Rank (std)        | 52.09 (32.84)  | 48.7615 (29.99) |
> >   | Rule-emphasized Mean Rank (std) | 53.11 (34.86)  | 40.17 (31.31)   |
> >
> > Neither prompt strategy leads to a stable or consistently improved performance. Although the rule-emphasized prompt slightly improves Claude’s mean rank, the large variance across both models indicates that the improvement is not systematic. The few-shot CoT prompt shows similarly unstable behavior: sometimes performing better, sometimes worse, and failing to produce reliable gains.
> >
> > Overall, these results suggest that:
> > 1. Prompt engineering has only a limited and inconsistent effect on non-uniform filtration generation—occasionally helping in isolated cases but failing to deliver robust improvements across the dataset.
> > 2. The instability and large variance reflect deeper difficulty in maintaining long-range filtration context, especially in tasks requiring accurate tracking of H0/H1 transitions.
> >
> > ---
> > **W3:** No Traditional PH Tool Comparison: Omits performance contrasts between LLMs, full traditional PH pipelines, and hybrid pipelines.
> >
> > **A3:**  Thank you for this helpful suggestion. In response, we have added a full comparison between LLM-based pipelines, hybrid pipelines, and a purely traditional PH pipeline. This allows us to directly measure the performance gap between LLM reasoning, PH Tools, and their combinations.
> >
> > In **Page 9** Section 3.5, we originally reported hybrid pipelines where parts of the PH workflow (e.g., filtration selection or diagram generation) are controlled by LLMs while the remaining steps use PH Tools. To address the reviewer’s comment, we now include configurations where the entire pipeline is controlled by PH Tools, except for the classifier, and filtration selection is no longer produced by the LLM. Instead, we use fixed filtrations (Weight, Degree, Betweenness, Closeness), providing a clear baseline for traditional PH computation.
> >
> > The updated comparison is shown below:
> >
> > | Filtration selection | Filtration values | Diagram generation | Distance comparison | Classifier | GPT-4o | GPT-4.1 | Claude | DS-V3 | Mistral |
> > |---|--|-|-|--|-|-|-|-|-|
> > | LLM| CODE| LLM | LLM | CODE| 0.150  | 0.180   | 0.780  | 0.290  | 0.210   |
> > | LLM| CODE| CODE| CODE| CODE| 0.350  | 0.350   | 0.350  | 0.350  | 0.350   |
> > | LLM | CODE| CODE| LLM| CODE | 0.250  | 0.280   | 0.840  | 0.780  | 0.520   |
> > | LLM| CODE| LLM| CODE| CODE | 0.300  | 0.320   | 0.190  | 0.025  | 0.150   |
> > | Weight (PH Tool)  | CODE| CODE| CODE| CODE |       0.790  | | | |  |
> > | Degree (PH Tool) | CODE| CODE| CODE| CODE       | 0.610  | | | | |
> > | Betweenness (PH Tool)| CODE| CODE| CODE| CODE| 0.890  | | | | |
> > | Closeness (PH Tool)  | CODE| CODE| CODE| CODE| 0.800  | | | | |
> >
> > The fully tool-based baselines significantly outperform all LLM-based and hybrid pipelines, which confirms that persistent homology computation is still far beyond the current abilities of LLMs. However, the hybrid setting shows that LLMs can still contribute meaningfully: for example, Claude achieves **0.840 accuracy** in one mixed configuration, even though it does not surpass the best tool-only baselines. This demonstrates that while LLMs struggle with the entire PH workflow, they can occasionally assist specific steps in a controlled pipeline.

---

> > > ### Author Response · Authors · 2025-11-21
> > >
> > > **W4:** Lacks key works on simplicial complex identification (e.g., Hypergraph Isomorphism Computation, Reinterpreting Hypergraph Kernels: Insights Through Homomorphism Analysis).
> > >
> > > **A4:** Thank you for pointing out these relevant works. We appreciate the suggestion and agree that research on hypergraph isomorphism computation and hypergraph kernel reinterpretation provides important perspectives on the identification and comparison of simplicial complexes. In the revision we have added a dedicated paragraph in the Related Work section that discusses these approaches and clarifies how they relate to persistent homology. **In page 14 you can find the paragraph that we have added:**
> > >
> > > Recent efforts in understanding and identifying higher-order topological structures have extended classical graph methods to the realm of hypergraphs and simplicial complexes. \[1\] analyzed the complexity of hypergraph isomorphism, showing that while the general problem is NP-complete, isomorphism between bounded-rank hypergraphs, such as low-dimensional simplicial complexes, admits moderately exponential-time algorithms via group-theoretic methods. To capture structural similarity beyond exact isomorphism, recent work has introduced polynomial-time heuristics based on combinatorial refinement. \[2\] proposed a generalization of the Weisfeiler–Lehman (WL) test to hypergraphs and constructed WL-based hypergraph kernels, enabling efficient comparisons between high-order structures through subtree and hyperedge patterns. Building on this, \[3\] reinterpreted such kernels in terms of homomorphism counts, revealing that many existing methods implicitly focus on acyclic patterns and introducing the Subtree-Cycle Kernel to enhance expressiveness by capturing cyclic features. These approaches provide both theoretical insights and practical tools for identifying and comparing simplicial complexes through a combination of isomorphism heuristics and homomorphism-informed embeddings.
> > >
> > >    \[1\] Babai L, Codenotti P. Isomorhism of hypergraphs of low rank in moderately exponential time\[C\]//2008 49th Annual IEEE Symposium on Foundations of Computer Science. IEEE, 2008: 667-676.
> > >    \[2\] Feng Y, Han J, Ying S, et al. Hypergraph isomorphism computation\[J\]. IEEE Transactions on Pattern Analysis and Machine Intelligence, 2024, 46(5): 3880-3896.
> > >    \[3\] Zhang Y, Du S, Feng Y, et al. Reinterpreting Hypergraph Kernels: Insights Through Homomorphism Analysis\[J\]. IEEE Transactions on Pattern Analysis and Machine Intelligence, 2025\.

---

### Author Response · Authors · 2025-12-02
**Overall Rebuttal Summary**

Dear AC and SAC,

We are grateful to the reviewers for their detailed and constructive feedback on our submission. Based on their valuable suggestions, we have incorporated the following updates into the revised version. We believe these additions have significantly improved the quality of our work.

### What Our Work Does

- Benchmark construction: We design the LLM4PH by following the full persistent homology (PH) workflow and organizing evaluation into four difficulty levels, each containing multiple subtasks such as filtration reasoning, barcode interpretation, and topological distance comparison.

- Cross-task analysis: We conduct extensive analyses, including prompt ablations, post-training and scale transferability, compositional PH pipelines, and detailed error analysis, to systematically assess how LLMs generalize across tasks and where their limitations emerge.

### Positive Feedback from Reviewers

Across all reviewers, our work received consistent strengths as follows.

1. Novelty:

    **All** reviewers agreed that the study is novel and interesting, and that evaluating language models on persistent homology opens a new direction.

2. Task quality:

    **Reviewers 9E47**, **XKxG**, and **tqDU** noted that our benchmark tasks are well designed and clearly structured.

3. Experimental strength:

    **Reviewers 9E47**, **gB2w**, and **tqDU** emphasized that our experiments are thorough. **All** of them appreciated that we tested multiple mainstream open source and closed source models and provided well documented code.

4. Writing quality:

    **Reviewers 9E47**, **XKxG**, and **gB2w** commented that the paper is well written and easy to follow.

5. Detailed analysis:

    Section 3.5 includes prompt ablations, post training and scale transferability, compositional PH pipeline evaluations, and error analysis for non uniform filtration. **Reviewer XKxG** specifically mentioned the error analysis as particularly insightful.

### Reviewer Concerns and How We Addressed Them
1. Motivation

    **Reviewers XKxG** and **gB2w** asked for a clearer articulation of why a PH-oriented LLM benchmark is necessary.

    *Response:*

    We have expanded the motivation to emphasize two goals:

    (1) evaluating LLMs as topological reasoning assistants rather than numerical PH solvers, and

    (2) probing fundamental reasoning limitations of LLMs on compositional, multi-scale structural tasks.

2. Missing Core PH Subtasks

    **Reviewers 9E47** and **XKxG** requested more PH-specific sub-tasks.

    *Response:*

    We have added two new tasks, i.e., Persistence Diagram Interpretation and Topological Distance Reasoning, with corresponding results.

3. Need for More Prompt Ablations

    **Reviewers 9E47**, **XKxG**, and **tqDU** asked for deeper analysis of how terminology and prompt formulation affect performance.

    *Response:*

    (1) We have added a terminology ablation experiment comparing topological vs. graph-theoretic phrasing, showing task-dependent sensitivity.

    (2) We have also added prompt-engineering experiments (Few-shot CoT & rule-emphasized prompting).

4. Hybrid Pipeline

    **Reviewers 9E47** and **tqDU** were very interested in hybrid LLM + PH tool approaches.

    *Response:*

    We have expanded Section 3.5 with new configurations where PH tools control the entire pipeline except filtration selection. We further introduced an additional optimization-based hybrid reasoning experiment where LLMs iteratively refine filtration strategies guided by PH computations.

5. Graph Isomorphism

    **Reviewer** **9E47** requested additional literature; **Reviewer** **tqDU** requested additional experiments.

    *Response:*

    (1) We have added references on simplicial complex identification and hypergraph kernels.

    (2) We have also added a new experiment using the CYCLES dataset to test whether LLMs can distinguish WL-indistinguishable graph pairs via PH reasoning.

6. More Advanced Architecture (e.g., GraphRAG, hierarchical structures)

    **Reviewer** **tqDU** asked whether better architectures would fix failures in multi-step topological reasoning.

    *Response:*

    We have designed a new memory-based hierarchical reasoning experiment, inspired by agent architectures, to address fact forgetting and rule forgetting. Results show partial improvements but persistent failures, demonstrating that limitations stem from model reasoning ability rather than prompting style alone.

### Conclusion
We sincerely thank all reviewers and the area chair for their thoughtful feedback. **Reviewer** **XKxG** raised their score (**from 4 to 6**) after the rebuttal, and **Reviewer** **tqDU** engaged actively with multiple follow-up questions. Although we could not see a final reply from **Reviewer** **tqDU**, we appreciate the constructive nature of their comments. We believe our extensive additions and new experiments have substantially strengthened the paper and clarified its contributions.

---

### Meta-Review · Area_Chair_TmeW · 2025-12-28

**Summary:**

This paper introduces LLM4PH, the first benchmark for evaluating LLMs' ability to reason about persistent homology on graphs. The benchmark decomposes the PH pipeline into four difficulty levels with multiple subtasks, testing models on tasks ranging from basic structural understanding to real-world graph inference. While all reviewers acknowledged the novelty of the direction and the thoroughness of experiments, concerns centered on motivation/applicability (why PH reasoning matters for LLMs).
The AC acknowledges that evaluating PH reasoning is intriguing and that the benchmark appears to be a valuable contribution. However, the AC is uncertain whether this contribution is substantial enough to warrant publication in ICLR.

**Reviewer Concerns:**

The rebuttal effectively addressed several concerns: (1) Two new PH-specific subtasks (Persistence Diagram Interpretation, Topological Distance Reasoning) were added ; (2) Terminology ablation experiments showed performance gaps are not primarily due to unfamiliar jargon; (3) Hybrid pipeline comparisons with traditional PH tools were included; (4) A new WL-indistinguishable graph experiment was added. However, the core motivation concern raised by Reviewers XKxG and gB2w—why PH reasoning is broadly important for LLM evaluation—remains only partially addressed.

**Reviewer Scores:**

One reviewer increased his score from 4 to 6. I am unsure if other reviewers whould have also raised their scores.

---

### Decision · Program_Chairs · 2026-01-26

Reject